# Conformal Reliability: A New Evaluation Metric for Conditional Generation

Yachen Gao [1 2]  Xinwei Sun [3]  Yikai Wang [4]  Ye Shi [5]  Jingya Wang [5]  Jianfeng Feng [1 3]  Yanwei Fu [2 3]

## Abstract

Conditional generative models have recently achieved remarkable success in various applications. However, a suitable metric for evaluating the reliability of these models, which takes into account their inherent uncertainty, is still lacking. Existing metrics, which typically assess a single output, may fail to capture the variability or potential risks in generation. In this paper, we propose a novel evaluation metric called *reliability score* based on conformal prediction, which measures the worst-case performance within the prediction set at a pre-specified confidence level. However, computing this score is challenging due to the high-dimensional nature of the output space and the nonconvexity of both the metric function and the prediction set. To efficiently compute this score, we introduce Conformal ReLiability (CReL), a framework that can **(i)** construct the prediction set with desired coverage; and **(ii)** accurately optimize the reliability score within the constructed prediction set. We provide theoretical results on coverage and demonstrate empirically that our method produces more informative prediction sets than existing approaches. Experiments on synthetic data and the image-to-text and text-to-image tasks further demonstrate the interpretability of our new metric, and the validity and effectiveness of our computational framework. Source code can be found at https://ggc29.github.io/CReL/.

[1]Institute of Science and Technology for Brain-Inspired Intelligence, Fudan University, Shanghai, China [2]Shanghai Innovation Institute, Shanghai, China [3]School of Data Science, Fudan University, Shanghai, China [4]Nanyang Technological University, Singapore [5]School of Information Science and Technology, ShanghaiTech University, Shanghai, China. Correspondence to: Xinwei Sun <sunxinwei@fudan.edu.cn>, Yanwei Fu <yanweifu@fudan.edu.cn>.

*Proceedings of the $43^{rd}$ International Conference on Machine Learning*, Seoul, South Korea. PMLR 306, 2026. Copyright 2026 by the author(s).

## 1. Introduction

Conditional generative models map a given input condition (*e.g.*, a textual prompt) to high-dimensional outputs (*e.g.*, images, sequences). Powered by large-scale datasets and models, this paradigm underpins breakthroughs in diverse domains, from text-to-image synthesis (Reed et al., 2016) and drug discovery (Bian & Xie, 2021) to autonomous systems (Gasparyan & Qiu, 2024). Yet, despite their remarkable generative prowess, a fundamental question remains largely unanswered: *how trustworthy are these models when deployed in the real world?*

Current metrics such as the CLIP score (Radford et al., 2021) typically assess a single generated output. While effective for measuring *average* capability, this paradigm fundamentally ignores the stochastic nature of generative models, masking potential risks hidden within the distribution of plausible outputs. A model may achieve a high average score by frequently generating high-quality samples, yet still harbor a non-negligible probability of producing catastrophic failures. For instance, in an image-to-text task, a model might correctly caption an image as "A man playing the guitar" in most cases, but under different sampling seeds, it could plausibly generate "A man pointing a gun" due to visual ambiguity or hallucinations. In safety-critical applications (Gawlikowski et al., 2023; He et al., 2026), reliability is defined not by how good the model *can* be, but by how bad it *might* be. Therefore, single-output evaluation is insufficient; a rigorous framework must quantify the *worst-case performance* among all statistically plausible outputs.

To provide such an assessment, we propose the *reliability score*, which evaluates the worst-case value of a user-specified similarity metric $\rho$ within a calibrated prediction set at confidence level $1 - \alpha$. In classical Conformal Prediction (CP), valid models are commonly compared using the geometric size (volume) of the prediction set—a notion of *sharpness*. However, in high-dimensional generative tasks, set volume is not only computationally intractable but also misaligned with the evaluation metric $\rho$: geometric size provides no information about how close the set remains to the ground truth under the metric of interest. To bridge this gap, our reliability score replaces geometric volume with a *metric-aware measure of sharpness*, defined as the worst achievable performance under $\rho$ among all statisti-

cally plausible outputs. This quantity directly reflects the model's performance *floor* at confidence level $1 - \alpha$. Yet, computing this score presents a significant challenge: directly optimizing the worst-case performance is intractable due to the high dimensionality of generative outputs, as well as the nonconvexity of both the similarity metric $\rho$ and the prediction set serving as the constraint.

For the regression task with multi-dimensional output, one can apply directional quantile regression (DQR) (Kong & Mizera, 2012; Paindaveine & Šiman, 2011), which can yield a convex prediction set. However, this set may be overly conservative, as it must account for extreme cases in order to guarantee coverage. Besides, the prediction set may not be informative since the true set may not be convex. Other methods (Xu et al., 2024; Javanmard et al., 2025; Feldman et al., 2023) leveraged conformal calibration, which can alleviate some of these issues. For instance, (Xu et al., 2024) modeled outputs as Gaussian mixtures or projected them into lower-dimensional latent spaces. Yet, none tackle the optimization of worst-case reliability under a general similarity metric—a problem that is both computationally and statistically intractable in high-dimensional output spaces.

To address these limitations, we introduce *Conformal ReLiability* (CReL), a principled computational framework for quantifying the reliability of conditional generative models. At its core, CReL projects high-dimensional outputs into a structured latent space, wherein both conformal calibration and optimization are performed. Compared to existing approaches that calibrate in the original output space, our method enjoys better computational efficiency and optimization tractability for computing the reliability score. Theoretically, we establish that the resulting prediction set satisfies the target guarantee. Moreover, reformulating the objective over the latent-space prediction set transforms the problem into an optimization program with convex constraints, on which the projection operation can be efficiently computed using linear programming. This formulation enables the employment of projected gradient descent, endowed with provable global convergence guarantees for computing the reliability score. We demonstrate the validity and effectiveness of our framework on synthetic data and on both the image-to-text and text-to-image tasks.

To summarize, our contributions are:

- *Reliability-Centric Metric*: We introduce the reliability score to quantify the worst-case performance of conditional generative models at a specified confidence level, addressing risks overlooked by single-sample evaluations.

- *CReL Framework*: we develop Conformal ReLiability (CReL), a computational framework that can efficiently and accurately compute the reliability score.

- *Theoretical Guarantee*: We show that the prediction set

generated by our method meets the coverage guarantee. Additionally, we empirically find that the prediction set given by our procedure has much smaller or comparable size to other methods, highlighting the effectiveness of our approach in delivering more informative calibration.

- *Empirical Validation*: We evaluate our methods on synthetic data and on both the image-to-text and text-to-image tasks. For synthetic data, we validate the effectiveness of our computational framework. In the image-to-text and text-to-image tasks, we demonstrate that our new metric provides more interpretable evaluations compared to traditional single-output metrics.

## 2. Related Works

**Evaluating condition generative models.** Typical metrics for evaluating conditional generative models include *Structural Similarity Index Measure* (SSIM) (Wang et al., 2004), *Contrastive Language-Image Pretraining* (CLIP) (Radford et al., 2021), and others. Specifically, SSIM evaluates the structural similarity between generated images and reference images, while CLIP measures the similarity between generated images and corresponding textual descriptions by projecting both modalities into a shared embedding space and calculating the cosine similarity between them. Other popular metrics, like BERT-similarity (Kenton & Toutanova, 2019) and *Fréchet Inception Distance* (FID) (Heusel et al., 2017), also rely on embedding models to quantify how well the generated samples match the given conditions. Recent studies have refined evaluation through conditional metrics (Benny et al., 2021), holistic benchmarks (Lee et al., 2023), and advanced diversity scores (Jalali et al., 2025; Ospanov et al., 2025). However, these methods focus primarily on average-case quality or aggregate diversity distributions. They do not capture *reliability*. While metrics like CLIP ignore variability and distributional scores obscure failure risks, our work fills this gap by explicitly quantifying the reliability of generative models.

**Conformal prediction for multi-dimensional data.** Many works have been done on this topic recently. (Kong & Mizera, 2012; Boček & Šiman, 2017) proposed directional quantile regression (DQR) that estimated quantile hyperplanes for multiple directions in the response space. However, this approach may remain conservative and uninformative, since the prediction set is constrained to be convex and requires estimating extreme quantiles to ensure coverage. While the vector quantile regression (Carlier et al., 2016) can produce non-convex sets, it restricts the output to linearly depend on the input. Other attempts include (Messoudi et al., 2022; Xu et al., 2024) that constructed the prediction set as an ellipsoidal set, (Wang et al., 2022b; Johnstone & Ndiaye, 2025; Gibbs et al., 2025; Plassier et al., 2025b) that modeled the conditional distribution of the output, and (Plassier

et al., 2025a) which rectifies scalar conformity scores via quantile regression to improve conditional coverage. In particular, (Feldman et al., 2023) proposed to map the high-dimensional response into a lower-dimensional latent space, which can alleviate the conservativeness problem when applying DQR. PCP (Wang et al., 2022b) is especially relevant for conditional generative models, but it is not designed for the reliability-score optimization considered here; its default multidimensional calibration can be more conservative than a joint latent-region calibration, as we empirically compare in Appendix C.4. While standard CP methods construct prediction sets, our goal is to evaluate a reliability score. This requires solving an optimization problem that is intractable under previous frameworks. For instance, (Feldman et al., 2023) generates non-convex prediction sets in the output space, rendering optimization infeasible. In contrast, our novelty lies in leveraging LGMs to construct a *convex* set within the latent space. This convexity ensures that optimizing the reliability score is computationally tractable.

# 3. Methodology

We aim to evaluate the reliability of a target model $f : \mathbb{R}^p \to \mathbb{R}^d$ in conditional regression tasks, with respect to a user-defined similarity metric $\rho$, where higher values of $\rho$ indicate better performance. Suppose our data has $n$ independent and identically distributed (*i.i.d.*) samples $\{(X_i, Y_i)\}_{i=1}^n$, where $X \in \mathbb{R}^p$ represents the input condition (*e.g.*, prompt) and $Y \in \mathbb{R}^d$ denotes the ground-truth output. For each $X_i$, the target model $f$ generates the output $\widehat{Y}_i := f(X_i)$.

Our goal is to assess the reliability of $f$ for a new observation $X_{n+1}$ with respect to $\rho$. Given a confidence level $\alpha \in (0, 1)$, we aim to quantify the worst-case performance at confidence level $1 - \alpha$:

$$\min_{\widehat{Y} \in C_{\mathcal{Y}}(X_{n+1})} \rho(\widehat{Y}, \mathrm{GT}_{n+1}),$$

$$\text{such that } \mathbb{P}\left(\widehat{Y}_{n+1} \in C_{\mathcal{Y}}(X_{n+1})\right) \geq 1 - \alpha, \quad (1)$$

where $\mathrm{GT}_{n+1}$ denotes the ground truth response, *i.e.*, $X_{n+1}$ (*e.g.*, CLIP similarity between the generated text and the image) or $Y_{n+1}$ (*e.g.*, BERT similarity between the true text and the generated image). This metric provides a robust, uncertainty-aware lower bound on performance, evaluating the reliability of the method in the worst-case allowed by the confidence level $1 - \alpha$.

Because $\widehat{Y}$ is normally high-dimensional, applying directional quantile regression (DQR) can result in overly conservative prediction sets, which may hinder an accurate assessment of reliability. To address this issue, we introduce *Conformal ReLiability* (CReL), a conformal framework built on the latent generative model, which allows both the calibration procedure and the optimization of $\rho(\cdot, \cdot)$ to be performed in a much lower-dimensional latent space

(as illustrated in Fig. 1). This approach yields more informative prediction sets and, consequently, more accurate reliability evaluation.

The rest of this section is organized as follows: Section 3.1 first introduces our conformal procedure on the latent space. Then, we will show in Section 3.2 that such a procedure can meet the coverage guarantee as long as the latent generative model is trained well. Finally, Section 3.3 introduces our optimization methods for computing the reliability score.

## 3.1. Conformal Calibration

The key insight of our framework is to learn an embedding space $\mathcal{Z}$ via a latent generative model, enabling conformal calibration in such a lower-dimensional space and thereby significantly reducing the over-conservativeness present in the original output space. Within this latent space, we construct $C_{\mathcal{Z}}(X_{n+1})$ through conformal calibration following DQR models (Kong & Mizera, 2012), transforming the original non-convex reliability optimization problem into a computationally tractable convex-constrained optimization.

We begin by partitioning the training indices into three folds: $\mathcal{I}_{\mathrm{lgm}}$ for training the latent generative model, $\mathcal{I}_{\mathrm{dqr}}$ for training the DQR model, and $\mathcal{I}_{\mathrm{cal}}$ for final conformal calibration. We denote the corresponding datasets as $\mathcal{D}_{\mathrm{lgm}}$, $\mathcal{D}_{\mathrm{dqr}}$, and $\mathcal{D}_{\mathrm{cal}}$, respectively.

**Step 1: Training the latent generative model.** The model is composed of an encoder that transforms $\widehat{Y}|X = x$ into a latent distribution $Z_x$ and constructs $C_{\mathcal{Z}}(X)$, followed by a decoder that gives the prediction set $C_{\mathcal{Y}}(X) := \mathcal{D}ec(C_{\mathcal{Z}}(X), X)$. It is ensured in Theorem 3.3 that $C_{\mathcal{Y}}(X)$ meets the coverage guarantee, as long as the latent generative model (LGM) can well recover the distribution $\widehat{Y}|X$. Typical choices of generative models satisfying this property include the *Variational Autoencoder* (VAE) (Khemakhem et al., 2020; Kingma & Welling, 2013) or the stable diffusion model (Rombach et al., 2022). To this end, we fit an LGM on $\{(X_i, \widehat{Y}_i)\}_{i \in \mathcal{I}_{\mathrm{lgm}}}$. After training, we can obtain an encoder $\mathcal{E}(\cdot, \cdot) : \widehat{\mathcal{Y}} \times \mathcal{X} \mapsto \mathcal{Z}$ and the decoder $\mathcal{D}ec(\cdot, \cdot) : \mathcal{Z} \times \mathcal{X} \mapsto \mathcal{Y}$. To align the encoder and decoder with the similarity metric $\rho$, we replace the mean square loss $\|\widehat{Y} - \mathcal{D}ec(\mathcal{E}(\widehat{Y}, X), X)\|_2^2$ with $\rho(\widehat{Y}, \mathcal{D}ec(\mathcal{E}(\widehat{Y}, X), X))$ during training. This LGM training is an offline preprocessing step and is empirically lightweight; detailed time and memory costs are reported in Appendix C.6.

**Step 2: Fitting the DQR model.** After training the LGM, we use the encoder $\mathcal{E}$ to obtain $(Z_i, X_i)$ from $(\widehat{Y}_i, X_i)$ for each $i \in \mathcal{I}_{\mathrm{dqr}}$, where $Z_i := \mathcal{E}(\widehat{Y}_i; X_i) \in \mathbb{R}^r$. Then, we apply the DQR (Kong & Mizera, 2012) on $\{Z_i, X_i\}_{i \in \mathcal{I}_{\mathrm{dqr}}}$ to obtain an initialized region $R_{\mathcal{Z}}(x)$ for any $x$ in the calibration dataset. Specifically, given a direction $\mathbf{u} \in \mathbb{S}^{r-1} := \{\mathbf{u} \in \mathbb{R}^r : \|\mathbf{u}\|_2 = 1\}$, DQR models the quantiles of a re-

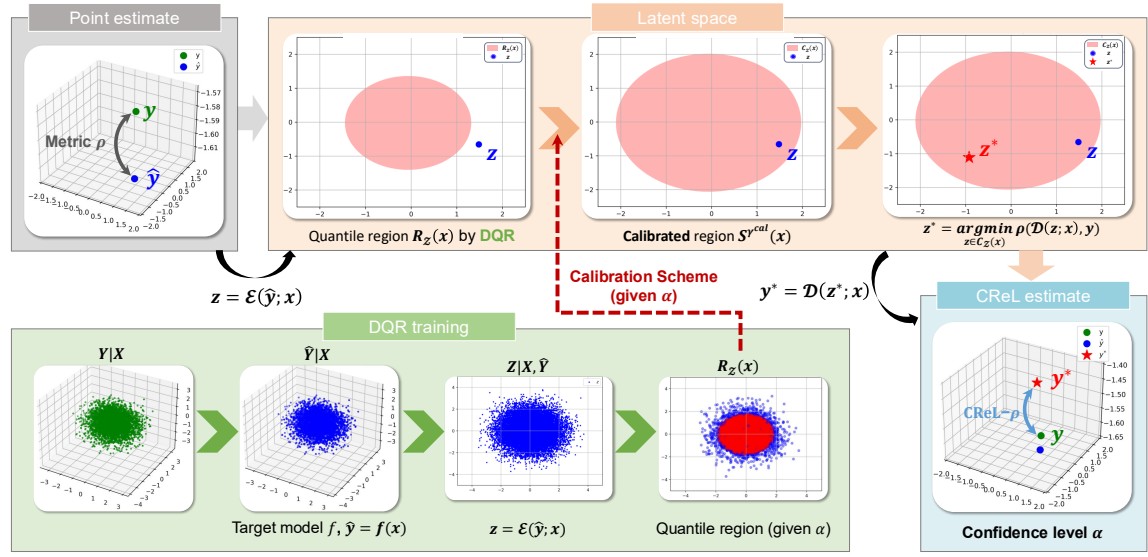

*Figure 1.* Illustration of our procedure. During DQR training, the latent generative model maps the target model's prediction space $\widehat{\mathcal{Y}}$ to a latent space $\mathcal{Z}$, where DQR constructs the quantile region $R_{\mathcal{Z}}(x)$ (3) using the loss (2). Then, CReL applies the calibration to adjust $R_{\mathcal{Z}}(x)$, such that the calibrated region $S^{\gamma_{\mathrm{cal}}}(x)$ satisfies the marginal coverage at level $1 - \alpha$. The final reliability metric CReL-$\rho$ is then computed by optimizing (8).

sponse vector in $\mathbf{u}$, allowing us to estimate the $\alpha$-th quantile for any input $X_i$ by projecting the response vector $Z_i$ onto $\mathbf{u}$. Specifically, DQR minimizes the following objective (2) to estimate the $\alpha$-th quantile of the projection $\mathbf{u}^\top Z_i$ given $X_i$:

$$\widehat{\beta} = \frac{1}{|\mathcal{D}_{\mathrm{dqr}}|} \sum_{i \in \mathcal{D}_{\mathrm{dqr}}} \zeta_\alpha \left( \mathbf{u}^\top Z_i, f_{\boldsymbol{\beta}}(X_i, \mathbf{u}) \right), \qquad (2)$$

where the pinball loss $\zeta_\alpha(y, \hat{y})$ is defined as:

$$\zeta_\alpha(y, \hat{y}) = \begin{cases} \alpha(y - \hat{y}) & \text{if } y - \hat{y} > 0, \\ (1 - \alpha)(\hat{y} - y) & \text{otherwise.} \end{cases}$$

Here, $f_{\boldsymbol{\beta}}(X_i, \mathbf{u})$ represents the regression function parameterized by $\boldsymbol{\beta}$, which predicts the value of the $\alpha$-th quantile for the projected data. For each direction $\mathbf{u}$, DQR defines a convex half-space $\mathbb{H}_u^+(x) = \left\{ z \in \mathbb{R}^r : \mathbf{u}^\top z \geq f_{\boldsymbol{\beta}}(x, \mathbf{u}) \right\}$. The quantile region is then obtained by taking the intersection of all such half-spaces across all directions $\mathbf{u} \in \mathbb{S}^{r-1}$:

$$R_{\mathcal{Z}}(x) = \bigcap_{\mathbf{u} \in \mathbb{S}^{r-1}} \mathbb{H}_u^+(x), \qquad (3)$$

which yields a convex region in the latent space $\mathcal{Z}$. The DQR fitting procedure is illustrated in Fig. 1, where the quantile region $R_{\mathcal{Z}}(x)$ constructed in $\mathcal{Z}$ during training is marked in red.

**Step 3: Calibration.** When the latent dimension $r > 1$, $R_{\mathcal{Z}}(X)$ covers strictly less than $1 - \alpha$ of the distribution, due to the intersection of the half-spaces. To address this, we perform calibration to construct $C_{\mathcal{Z}}(X)$ on $\mathcal{D}_{\mathrm{cal}} := \{(X_i, Z_i)\}_{i \in \mathcal{I}_{\mathrm{cal}}}$, such that $\mathbb{P}\left( Z_{n+1} \in C_{\mathcal{Z}}(X_{n+1}) \right) \geq 1 -$

$\alpha$. Here, DQR provides a convex latent base region, while the additional CReL calibration is applied jointly to the entire prediction region rather than coordinate by coordinate. This joint calibration targets not only validity but also informativeness, seeking a compact prediction set among those satisfying the desired coverage. To this end, we first define a base region:

$$S^\gamma(x) = \left\{ z \in \mathbb{R}^r : \min_{a \in R_{\mathcal{Z}}(x)} d(a, z) \leq \gamma \right\}, \qquad (4)$$

where $d(\cdot, \cdot)$ a distance function. The goal is to find a $\gamma_{\mathrm{cal}}$ such that $C_{\mathcal{Z}}(X) = S^{\gamma_{\mathrm{cal}}}(X)$. To achieve the target coverage, we expect $\gamma_{\mathrm{cal}}$ to be the $(1 - \alpha)$-quantile of the distribution function over $\min_{a \in R_{\mathcal{Z}}(X)} d(a, \cdot)$. We first obtain the coverage rate $\gamma_{\mathrm{init}}$ of the uncalibrated base regions (*i.e.*, $S^0(X_i) = R_{\mathcal{Z}}(X_i)$):

$$\gamma_{\mathrm{init}} = \frac{1}{|\mathcal{D}_{\mathrm{cal}}|} \left| \{ Z_i : Z_i \in R_{\mathcal{Z}}(X_i), i \in \mathcal{I}_{\mathrm{cal}} \right|, \qquad (5)$$

where $Z_i = \mathcal{E}(\widehat{Y}_i; X_i)$ for each $i \in \mathcal{I}_{\mathrm{cal}}$. Since the coverage of $R_{\mathcal{Z}}(X_i)$ is strictly less than $1 - \alpha$ when $r > 1$, we would have $\gamma_{\mathrm{init}} \leq 1 - \alpha$ as long as the sample size of the calibration data, *i.e.*, $|\mathcal{I}_{\mathrm{cal}}|$ is sufficiently large. That means, to achieve coverage guarantee, we should grow the base quantile region by computing $\gamma_{\mathrm{cal}}$ as

$$E_i^+ = \min_{a \in R_{\mathcal{Z}}(X_i)} d(a, Z_i), \quad \forall i \in \mathcal{I}_{\mathrm{cal}},$$
$$\gamma_{\mathrm{cal}} := \lceil (|\mathcal{D}_{\mathrm{cal}}| + 1)(1 - \alpha) \rceil\text{-th smallest} \qquad (6)$$
$$\text{value of } \{ E_i^+ : i \in \mathcal{I}_{\mathrm{cal}} \}.$$

Finally, the calibrated quantile region $C_{\mathcal{Z}}(X)$ is given by $S^{\gamma_{\mathrm{cal}}}(X)$ in (4).

**Algorithm 1** Conformal ReLiability

---

**Input:** Dataset $\{(X_i, Y_i)\}_{i=1}^n$, target model $f$, similarity metric $\rho$, nominal confidence level $\alpha \in (0, 1)$, encoder $\mathcal{E}(\cdot, \cdot)$ and decoder $\mathcal{D}ec(\cdot, \cdot)$ in the latent generative model, DQR algorithm, a test point $X_{n+1}$.

**Output:** $C_{\mathcal{Y}}(X_{n+1})$.

**Training time:**
1: Split $\{1, \cdots, n\}$ into three disjoint sets $\mathcal{I}_{\mathrm{lgm}}$, $\mathcal{I}_{\mathrm{dqr}}$, $\mathcal{I}_{\mathrm{cal}}$.
2: Train a latent generative model on $\{(X_i, \widehat{Y}_i)\}_{i \in \mathcal{I}_{\mathrm{lgm}}}$, where $\widehat{Y}_i := f(X_i)$ for each $i = 1, ..., n$.
3: Fit a DQR model on $\{(X_i, Z_i)\}_{i \in \mathcal{I}_{\mathrm{dqr}}}$, where $Z_i := \mathcal{E}(\widehat{Y}_i; X_i)$, and to obtain $R_{\mathcal{Z}}(x)$ (3).

**Calibrating time:**
1: Compute the coverage of the uncalibrated quantile regions on $\{(X_i, Z_i)\}_{i \in \mathcal{I}_{\mathrm{cal}}}$ via (5).
2: Compute $E_i^+$ and obtain $\gamma_{\mathrm{cal}}$ according to (6).

**Test time:**
1: Obtain a base quantile region $R_{\mathcal{Z}}(X_{n+1})$ using a pre-trained DQR model.
2: Construct the calibrated quantile region $S^{\gamma_{\mathrm{cal}}}(X_{n+1})$ according to (4).
3: Construct $C_{\mathcal{Y}}(X_{n+1}) = \mathcal{D}ec(S^{\gamma_{\mathrm{cal}}}(X_{n+1}), X_{n+1})$.

---

*Remark* 3.1. Unlike (Feldman et al., 2023), which performs calibration in the output space $\mathcal{Y}$, we calibrate directly in the latent space $\mathcal{Z}$. This is motivated by computational and optimization considerations. Specifically, (Feldman et al., 2023) calibrates on $R_{\mathcal{Y}}(X) = \mathcal{D}ec(R_{\mathcal{Z}}(X), X)$. Since $R_{\mathcal{Y}}(X)$ may be non-convex, calibration requires discretization, which can be computationally expensive, particularly in high-dimensional spaces. In contrast, $R_{\mathcal{Z}}(X)$ is convex, allowing the core $E_i^+$ to be computed efficiently via linear programming. Please refer to Appendix D for more details about computational complexity. Furthermore, direct optimizing (1) over $C_{\mathcal{Y}}(X_{n+1})$ can be intractable, as both $C_{\mathcal{Y}}(X_{n+1})$ and $\rho(\cdot, \cdot)$ are non-convex. In contrast, as we will show, the optimization can be reformulated into one that optimizes in the latent space $C_{\mathcal{Z}}(X_{n+1}) = S^{\gamma_{\mathrm{cal}}}(X_{n+1})$. Because this space is convex and compact, the optimization becomes more tractable.

**Step 4: Constructing $C_{\mathcal{Y}}(X_{n+1})$.** Our final prediction set is given by $C_{\mathcal{Y}}(X_{n+1}) := \mathcal{D}ec(S^{\gamma_{\mathrm{cal}}}(X_{n+1}), X_{n+1})$. Alg. 1 summarizes the overall procedure for calibration.

### 3.2. Theoretical Guarantee

The role of our theoretical results is to formally support the proposed reliability metric and latent-space CReL framework. Proposition 3.2 establishes coverage after latent-space calibration, Theorem 3.3 shows that the decoded prediction set inherits the desired coverage guarantee, and Proposition 3.5 identifies the convex and compact structure that makes projected optimization tractable.

In this section, we provide the coverage guarantee for $C_{\mathcal{Y}}(X_{n+1})$. First, we show that after calibration,

$C_{\mathcal{Z}}(X_{n+1}) = S^{\gamma_{\mathrm{cal}}}(X_{n+1})$ satisfies the coverage guarantee in the latent space.

**Proposition 3.2.** *Suppose data in $\mathcal{D}_{\mathrm{lgm}}$, $\mathcal{D}_{\mathrm{dqr}}$, and $\mathcal{D}_{\mathrm{cal}} \cup \{X_{n+1}, Y_{n+1}\}$ are independent to each other. Besides, we assume $\{X_i, Y_i\}_{i \in \mathcal{I}_{\mathrm{cal}}} \cup (X_{n+1}, Y_{n+1})$ are exchangeable. Given a nominal coverage level $\alpha \in (0, 1)$, the quantile region $S^{\gamma_{\mathrm{cal}}}(X_{n+1})$ given by Alg. 1 satisfies:*

$$1 - \alpha \leq \mathbb{P}\left(Z_{n+1} \in S^{\gamma_{\mathrm{cal}}}(X_{n+1})\right) \leq 1 - \alpha + \frac{1}{1 + |\mathcal{D}_{\mathrm{cal}}|}.$$

*Proof.* The goal is to show that $\{E_i^+\}_{i \in \mathcal{I}_{\mathrm{cal}}} \cup E_{n+1}^+$ are exchangeable. First, since $f$ has been trained and is fixed, $\{(X_i, \widehat{Y}_i)\}_{i \in \mathcal{I}_{\mathrm{cal}} \cup \{n+1\}}$ are exchangeable. Since for each $i \in \mathcal{I}_{\mathrm{cal}}$, $Z_i$ is obtained from $\mathcal{E}(\widehat{Y}_i, X_i)$, and the encoder $\mathcal{E}(\cdot, \cdot)$ is trained on $\mathcal{D}_{\mathrm{lgm}}$ that are independent to $\mathcal{D}_{\mathrm{cal}}$, we have $\{(X_i, Z_i)\}_{i \in \mathcal{I}_{\mathrm{cal}} \cup \{n+1\}}$ are exchangeable. Since $E_i^+$ for each $i \in \mathcal{I}_{\mathrm{cal}}$ is determined by $\mathcal{D}_{\mathrm{dqr}}$ that are independent to $\mathcal{D}_{\mathrm{cal}} \cup \{X_{n+1}, Y_{n+1}\}$, $\{E_i^+\}_{i \in \mathcal{I}_{\mathrm{cal}}} \cup E_{n+1}^+$ are exchangeable. $\square$

Using this property, we can further demonstrate that, provided the latent generative model accurately recovers the conditional distribution $Y | X = x$, the resulting prediction set $C_{\mathcal{Y}}(X_{n+1})$ also satisfies the desired coverage.

**Theorem 3.3.** *Assume conditions in proposition 3.2 hold. Besides, we assume that $\forall x \in \mathcal{X}$, $\mathcal{D}ec(\mathcal{E}(\widehat{Y}, x), x) =_d \widehat{Y} | X = x$. Given any nominal coverage level $\alpha \in (0, 1)$, $C_{\mathcal{Y}}(X_{n+1}) := \mathcal{D}ec(S^{\gamma_{\mathrm{cal}}}(X_{n+1}), X_{n+1})$ given by Alg. 1 satisfies:*

$$\mathbb{P}\left(\widehat{Y}_{n+1} \in C_{\mathcal{Y}}(X_{n+1})\right) \geq 1 - \alpha.$$

*Proof.* First, by proposition 3.2, we have $\mathbb{P}(Z_{n+1} \in S^{\gamma_{\mathrm{cal}}}(X_{n+1})) \geq 1 - \alpha$. Since $Z_{n+1} \in S^{\gamma_{\mathrm{cal}}}(X_{n+1}) \implies \mathcal{D}ec(Z_{n+1}, X_{n+1}) \in \mathcal{D}ec(S^{\gamma_{\mathrm{cal}}}(X_{n+1}), X_{n+1})$, we have:

$$\mathbb{P}\left(\mathcal{D}ec(Z_{n+1}, X_{n+1}) \in \mathcal{D}ec(S^{\gamma_{\mathrm{cal}}}(X_{n+1}), X_{n+1})\right)$$
$$\overset{(1)}{\geq} \mathbb{P}(Z_{n+1} \in S^{\gamma_{\mathrm{cal}}}(X_{n+1})) \geq 1 - \alpha. \tag{7}$$

Since $\mathcal{D}ec(\mathcal{E}(\widehat{Y}, x), x) =_d \widehat{Y} | X = x$, we further have

$$\mathbb{P}(\widehat{Y}_{n+1} \in C_{\mathcal{Y}}(X_{n+1}))$$
$$= \mathbb{P}\left(\mathcal{D}ec(\mathcal{E}(\widehat{Y}_{n+1}, X_{n+1}), X_{n+1}) \in C_{\mathcal{Y}}(X_{n+1})\right)$$
$$\geq 1 - \alpha.$$

This completes the proof. $\square$

*Remark* 3.4. Compared to (Feldman et al., 2023), which performs calibration directly in the original output space $\mathcal{Y}$, the prediction set $C_{\mathcal{Y}}(X_{n+1})$ generated by our method may be slightly more conservative in terms of coverage. This is

due to the effect described in "(1)" of (7), where the decoder can expand the region. Nevertheless, this slight increase in conservativeness is a worthwhile trade-off, as it facilitates optimization when computing reliability scores. Moreover, as demonstrated empirically, the degree of overconservativeness is minor, *i.e.*, the size of the resulting region is comparable to that reported in (Feldman et al., 2023).

The assumption that LGM can well recover the conditional distribution has been similarly made in (Feldman et al., 2023). This property can hold for many types of latent generative models, including the variational autoencoder (Khemakhem et al., 2020), and the stable diffusion model (Rombach et al., 2022; Li et al., 2023). We further examine this assumption in Appendix C.5 through a reconstruction-error sensitivity analysis: when the VAE reconstruction loss is sufficiently small, coverage remains stable near the target level, whereas large reconstruction error leads to coverage failure. This suggests that a properly trained LGM mitigates the risk of evaluation bias induced by imperfect reconstruction.

### 3.3. Optimization

After constructing $C_{\mathcal{Z}}(X_{n+1}) := S^{\gamma_{\text{cal}}}(X_{n+1})$ and $C_{\mathcal{Y}}(X_{n+1})$, we are ready to compute the reliability (1) given a metric $\rho$. Since $C_{\mathcal{Y}}(X_{n+1}) := \mathcal{D}ec(C_{\mathcal{Z}}(X_{n+1}), X_{n+1})$, it is equivalent to consider the following objective:

$$\min_{z \in C_{\mathcal{Z}}(X_{n+1})} \rho\left(\mathcal{D}ec(z; X_{n+1}), \text{GT}_{n+1}\right). \quad (8)$$

Compared with the original objective, where both $\rho$ and the constraint set may be nonconvex, the feasible region $C_{\mathcal{Z}}(X_{n+1})$ is convex and compact, as shown below. Thus objective (8) falls into the category of nonconvex optimization over a convex set, as studied in (Lacoste-Julien, 2016).

**Proposition 3.5.** *If $R_{\mathcal{Z}}(x)$ is convex and $d(\cdot, \cdot)$ is jointly convex, $S^{\gamma}(x)$ is a convex and compact set for any $\gamma$.*

*Remark* 3.6. The joint convexity can hold for any norm-induced distance (*i.e.*, $d(a, b) := \|a - b\|$), Bregman divergence, or $f$-divergence. In this paper, we choose $d(a, b)$ to be the Euclidean distance.

Moreover, by (6), it is easy to see that the projection onto $S^{\gamma}(x)$ can be efficiently solved via a linear programming algorithm. Specifically, suppose $d(x, y) := \|x - y\|_2$. To compute $\Pi_{S^{\gamma_{\text{cal}}}(x)}(y) := \arg\min_{z \in S^{\gamma_{\text{cal}}}(x)} \|y - z\|_2$ given a new point $y$ to be projected, we can see that:

$$\Pi_{S^{\gamma_{\text{cal}}}(x)}(y) = \begin{cases} y & \text{if } y \in S^{\gamma_{\text{cal}}}(x) \\ y^* + \gamma_{\text{cal}} \frac{y^* - y}{\|y^* - y\|_2} & \text{otherwise,} \end{cases}$$

where $y^* := \arg\min_{y_1 \in R_{\mathcal{Z}}(x)} \|y_1 - y\|_2$. It is then sufficient to compute $\Pi_{R_{\mathcal{Z}}(x)}(y)$, which can be formulated as

the linear programming problem as follows:

$$\min_{y_1} \|y_1 - y\|_2^2,$$
$$\text{s.t. } \mathbf{u}_k^\top y_1 \geq f_{\widehat{\beta}}(x, \mathbf{u}_k), \quad k = 1, ..., K,$$

where $\widehat{\beta}$ is obtained after DQR training, and $R_{\mathcal{Z}}(x)$ is constructed using $K$ directions $\mathbf{u}_1, ..., \mathbf{u}_K$. As the projection can be efficiently computed, we can implement projected gradient descent (Ghadimi et al., 2016; Ghadimi & Lan, 2016), whose global convergence property has been established (Ghadimi & Lan, 2016). To find the global optimal, we use random search to pick several starting points and then apply the projected gradient descent to the initial that has the smallest $\rho$. Empirically, this procedure is stable to random initialization: on image-to-text evaluation with BLIP-base and CLIP-SIM, using $\alpha = 0.1$ and $\text{num}_{z_0} = 50$, the standard deviation across 10 repeated runs is only 0.00027; see Appendix C.8.

## 4. Experiments

### 4.1. Experiments on Synthetic Datasets

**Setups.** We consider the *nonlinear* synthetic setting (see Appendix B.1 for generation details). We generate $50,000$ samples and set $p = 38$, $d = 2$, and $\epsilon = 0.3$. The dataset is split as follows: 60% for training the latent generative model, 24% for training the DQR, 8% for calibration, and 8% for testing. By default, we adopt the Mean Squared Error (MSE) as the similarity metric $\rho$, consistent with the standard VAE reconstruction loss. We report the average coverage ratios of $C_{\mathcal{Z}}(X)$ and $C_{\mathcal{Y}}(X)$, as well as the area (defined as the number of grid points falling into the region) of the calibration region $C_{\mathcal{Y}}(X)$. For comparison, we also report the coverage ratios and areas of the calibration regions for the method of (Feldman et al., 2023) and the standard DQR method. Additionally, to demonstrate that our coverage guarantees are robust to the choice of metric, we provide results using the Mean Absolute Error (MAE) in Appendix C.3.

**Implementation details.** We set the latent space dimension to $r = 2$. For the latent generative model, we choose VAE and set the KL regularization hyperparameter $\beta = 0.001$ (see Appendix C for the ablation study on the choice of $r$ and $\beta$). For DQR in our method and in Feldman's method, the input size is $p + d$, and each gradient step uses $1,024$ directions with $\alpha = 0.1$. All data are $L_2$-normalized before training. Since setting the quantile level to $\alpha$, DQR does not achieve the target coverage, we decrease the quantile level until the target coverage is met, resulting in the quantile levels of 0.01 and 0.001 to achieve coverage rates of $1 - \alpha = 0.9$ and 0.98, respectively. For simplicity, we denote them as DQR-0.01 and DQR-0.001. More details can be found in Appendix B.2.

*Table 1.* Coverage ratio and area on the *nonlinear* synthetic dataset with different nominal levels $\alpha$.

| $\alpha$ | Coverage | | | | | Area in $\mathcal{Y}$ | | |
|---|---|---|---|---|---|---|---|---|
| | Ours-$\mathcal{Z}$ | Ours-$\mathcal{Y}$ | Feldman-$\mathcal{Y}$ | DQR-$\mathcal{Z}$ | DQR-$\mathcal{Y}$ | Ours | Feldman | DQR |
| 0.02 | 0.9770 | 0.9760 | 0.9718 | 0.9818 | 0.9872 | 398.5 | 377.8 | 749.1 |
| 0.10 | 0.8953 | 0.8915 | 0.8940 | 0.8823 | 0.9145 | 232.7 | 234.5 | 287.4 |

**Calibration result.** As shown in Table 1, all methods achieve the target coverage, and the prediction set region of our method is much smaller than that of DQR. This shows that, although both DQR and CReL approximately attain the desired coverage, the proposed joint calibration yields a more informative prediction set. We also note that the region is slightly larger than that of Feldman, which may be due to the expansion caused by the decoder (7). We also compare with PCP (Wang et al., 2022b) on the nonlinear simulation with $\alpha = 0.1$: PCP obtains coverage 0.9001 with area 854.24, whereas CReL obtains coverage 0.8915 with area 232.70; see Appendix C.4 for settings.

**Visualization.** We visualize the region for two different values of $X$, where $R_{\mathcal{Z}}$ represents the region before calibration (4), and $R_{\mathcal{Y}}$ denotes the decoded region of $R_{\mathcal{Z}}$, *i.e.*, $R_{\mathcal{Y}} = \mathcal{D}ec(R_{\mathcal{Z}}, x)$. As shown in Fig. 2, the pre-calibrated region (in red) initially excludes the outcome (in green), but after calibration, the outcome is successfully included. Compared to DQR, the regions produced by our method and Feldman's approach are smaller. Additionally, the regions have different shapes across the two cases, demonstrating the adaptiveness of our calibration procedure.

**CReL scales efficiently for high-dimensional calibration.** We evaluate the total calibration runtime across latent dimensions and find that our method scales efficiently with $r$, while the grid-based approach (Feldman et al., 2023) incurs exponential growth and becomes infeasible in high dimensions (see Appendix E). This confirms CReL's practicality for modern large-scale systems.

### 4.2. Experiments on Image-to-Text Task

**Dataset and preprocessing.** We use the MS-COCO 2014 validation set (Lin et al., 2014) ($40,504$ image-caption pairs), and split it into $75\%$ for VAE training, $15\%$ for DQR, $5\%$ for calibration, and $5\%$ for testing. We evaluate four models: BLIP (base and large) (Li et al., 2022) and GIT (base and large) (Wang et al., 2022a), all at image size $224 \times 224$. We measure CLIP cosine similarity (CLIP-SIM) between the condition image and generated caption, *i.e.*, $\rho(\widehat{Y}, \text{GT}) = \text{Cos-Sim}(\widehat{Y}, X)$, where $X$ denotes the image feature and $\widehat{Y}$ denotes the generated caption. We also measure the BERT cosine similarity (BERT-SIM) between the ground truth caption and generated caption, *i.e.*, $\rho(\widehat{Y}, \text{GT}) = \text{Cos-Sim}(\widehat{Y}, Y)$, which first extracts features from $\widehat{Y}$ and $Y$ and then computes the cosine similarity between these feature representations.

*Table 2.* Quantitative results of the image-to-text generation task at $\alpha = 0.1$, with differences between CReL-$\rho$ and $\rho$ ($\Delta$) highlighted in blue. Superscripts indicate the performance rank.

| Model | CLIP-SIM | | BERT-SIM | |
|---|---|---|---|---|
| | CLIP | CReL-CLIP | BERT | CReL-BERT |
| BLIP-base | $0.2330^4$ | $0.0070^1$ $(-0.2260)$ | $0.8349^3$ | $0.6335^3$ $(-0.2014)$ |
| BLIP-large | $0.2453^3$ | $-0.0074^4$ $(-0.2527)$ | $0.8106^4$ | $0.5631^4$ $(-0.2475)$ |
| GIT-base | $0.2511^2$ | $-0.0021^2$ $(-0.2532)$ | $0.8620^2$ | $0.6474^1$ $(-0.2146)$ |
| GIT-large | $0.2550^1$ | $-0.0043^3$ $(-0.2593)$ | $0.8649^1$ | $0.6459^2$ $(-0.2190)$ |

**Implementation details.** Image features are extracted using CLIP ViT-L/14 (Radford et al., 2021), and caption features are obtained using BERT-base (Kenton & Toutanova, 2019), where we use the [CLS] token that serves as a summary feature of the entire caption. Both feature types have dimension $p = d = 768$. The maximum text length is set to 50. For VAE, we set $r = 50$ and use $\beta = 0.001$ for KL regularization (see Appendix C for ablation study). For DQR, the input size is $p + d$, and each gradient step uses $1,024$ directions with $\alpha = 0.1$. All data are $L_2$-normalized before training. During optimization, we initialize the procedure with 50 starting points for BERT and CLIP. More details can be found in Appendix C.8.

**Quantitative comparison.** We compare two large-scale caption generation models (BLIP and GIT) in both base and large versions, using two similarity metrics: CLIP-SIM and BERT-SIM. Results at $\alpha = 0.1$ is shown in Tab. 2. For CLIP-SIM, the rankings among BLIP-base, BLIP-large, and GIT-large vary after calibration. Notably, BLIP-base ranks last in the original score but first in our reliability score. This can be explained by Fig. 3(a), where the distribution of BLIP-base's scores is more concentrated around the central region compared to GIT-large, resulting in a higher worst-case score after calibration. For BERT-SIM, we observe that the gap between BLIP-base and BLIP-large is enlarged after calibration. Similarly, this can be explained by the more concentrated score distribution of BLIP-base relative to BLIP-large. In addition, our results also indicate that BLIP-base is the most reliable one in CLIP-SIM, while the GIT-base/large achieves the highest reliability in BERT-similarity. This may be because CLIP-SIM captures high-level semantic similarity between generated image and text, making it more suited to lightweight models like BLIP-base that avoid overfitting to irrelevant features. In contrast, BERT-SIM focuses on deeper and subtler contextual similarity. As a result, the GIT model, with its larger capacity and ability to process more intricate relationships, performs better in this task.

**Qualitative results: CReL effectively identifies misalignments.** Figure 4 presents examples illustrating that our calibrated metric better reflects generation quality compared to the original uncalibrated metric. Specifically, we take examples from CReL-CLIP (image–caption) and CReL-BERT (caption–caption). In the example for CReL-CLIP

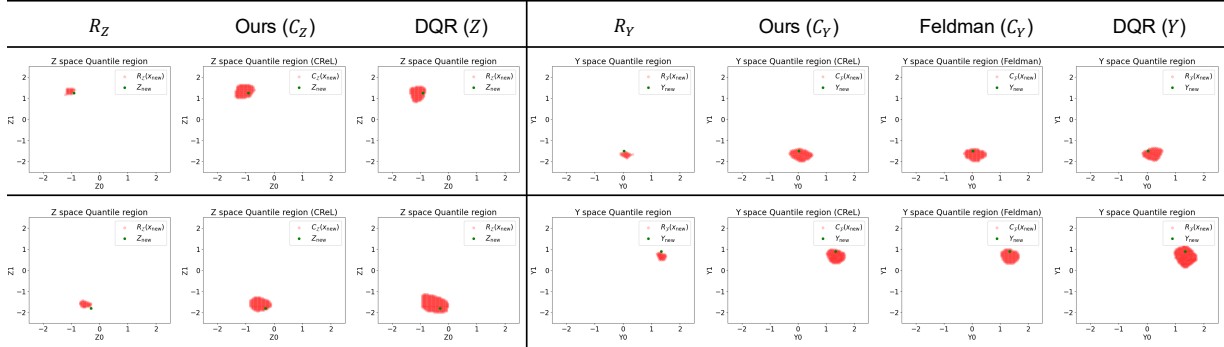

*Figure 2.* Visualization of regions produced by various methods ($\alpha = 0.1$). Each row represents a case, *i.e.*, a fixed $x$. **Left:** region in $\mathcal{Z}$; **Right:** region in $\mathcal{Y}$. Calibrated regions are marked in red, the test sample ($Z_{\text{new}}$ or $Y_{\text{new}}$) is marked in green.

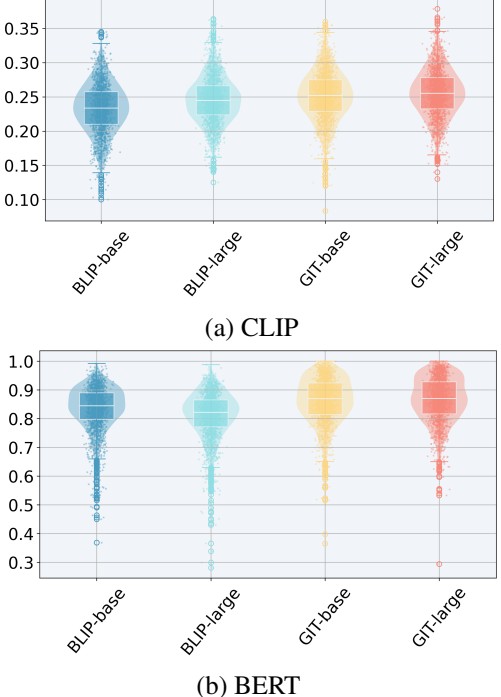

(a) CLIP

(b) BERT

*Figure 3.* Distribution of $\rho$ values for different models on the image-to-text generation task.

(*i.e.*, image-caption), the ground truth image shows "a baby in a seat playing a toy", but the GIT-base overlooks the information of "playing a toy". Despite this omission, CLIP assigns higher similarity scores to the GIT-base than the BLIP-large, which correctly identifies this semantics and is accurately ranked first by our reliability metric. In the example for CReL-BERT (*i.e.*, caption-caption), both BLIP-base ("a group of cell phones on a table") and GIT-large ("three phones sitting on a table") fail to specify the number of phones or mention that the table surface is wooden; however, they are ranked higher than BLIP-large, whose caption accurately captures both pieces of information. These results demonstrate that CReL effectively detects visual and semantic discrepancies that standard metrics miss, quantifying model reliability without sacrificing predictive performance.

*Table 3.* Quantitative results of the text-to-image generation task at $\alpha = 0.1$ under CLIP-SIM and DINO-SIM. Differences between CReL scores and average scores ($\Delta$) are shown in parentheses. Superscripts indicate the performance rank.

| Model | CLIP-SIM | | DINO-SIM | |
|---|---|---|---|---|
| | CLIP | CReL-CLIP | DINO | CReL-DINO |
| SD3-M | $0.2590^3$ | $0.0134^1$ $(-0.2456)$ | $0.4615^1$ | $-0.1480^4$ $(-0.6095)$ |
| SD3.5-L | $0.2596^2$ | $0.0116^2$ $(-0.2480)$ | $0.4531^2$ | $-0.1365^1$ $(-0.5896)$ |
| FLUX.1-dev | $0.2509^4$ | $0.0056^4$ $(-0.2453)$ | $0.4395^4$ | $-0.1411^3$ $(-0.5806)$ |
| Kandinsky-2.2 | $0.2603^1$ | $0.0062^3$ $(-0.2541)$ | $0.4407^3$ | $-0.1404^2$ $(-0.5811)$ |

More examples can be found in Appendix F.1.

### 4.3. Experiments on Text-to-Image Task

**Dataset and preprocessing.** We use the MS-COCO 2014 validation set (Lin et al., 2014) ($40,504$ image-caption pairs), and split it into $75\%$ for VAE training, $15\%$ for DQR, $5\%$ for calibration, and $5\%$ for testing. We evaluate four models: SD3-M (Esser et al., 2024), SD3.5-L (Esser et al., 2024), FLUX.1-dev (Labs, 2024), and Kandinsky-2.2 (Shakhmatov et al., 2023), all at image size $512 \times 512$ with 50 inference steps and guidance scale 7.0. We use two metrics: CLIP cosine similarity (CLIP-SIM), which measures caption-image semantic alignment using CLIP features, and DINO cosine similarity (DINO-SIM), which measures structural similarity between generated and ground-truth images using DINOv2-base features (Oquab et al., 2023). Please refer to Appendix F.2 for additional implementation details and results.

**Quantitative comparison.** We compare four text-to-image models under CLIP-SIM and DINO-SIM at $\alpha = 0.1$, with results in Tab. 3 and score distributions in Fig. 11. For CLIP-SIM, Fig. 11a shows that the four distributions largely overlap: Kandinsky-2.2 has the highest average CLIP score, but its distribution is not clearly separated from SD3-M or SD3.5-L. Thus, its average advantage is concentrated in the central/upper part of the distribution. CReL instead focuses on the lower-performance region inside the calibrated prediction set, where SD3-M outperforms the other methods. This

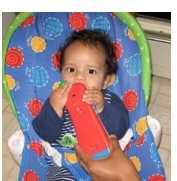

| | CLIP | CReL-CLIP |
|---|---|---|
| **GT caption:** A baby in a bouncy seat chewing on a plastic toy. | | |
| **BLIP-base:** a baby in a car seat | 0.1920[4] | -0.0202[2] |
| **BLIP-large:** there is a baby sitting in a high chair with a toy in his mouth | 0.2306[2] | -0.0039[1] |
| **GIT-base:** my son in his high chair | 0.2529[1] | -0.0443[4] |
| **GIT-large:** sitting in a chair with a red toy | 0.2213[3] | -0.0435[3] |

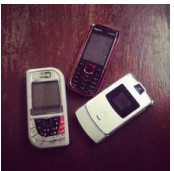

| | BERT | CReL-BERT |
|---|---|---|
| **GT caption:** Three cell phones lying next to each other on a wooden table. | | |
| **BLIP-base:** a group of cell phones sitting on a table | 0.8825[3] | 0.6388[4] |
| **BLIP-large:** three cell phones are sitting on a table with a wooden surface | 0.7509[4] | 0.6560[2] |
| **GIT-base:** three cell phones sitting on top of a wooden table. | 0.9880[1] | 0.6627[1] |
| **GIT-large:** three cell phones sitting on a table. | 0.9788[2] | 0.6421[3] |

*Figure 4.* Qualitative results of image-to-text models ($\alpha = 0.1$). Superscripts denote rank.

explains why SD3-M ranks first under CReL-CLIP despite ranking only third under average CLIP, while FLUX.1-dev remains last due to its lower central region and more visible low-score tail. For DINO-SIM, the distributions in Fig. 11b are much wider. SD3-M has a strong upper-score region, which raises its average DINO score, but it also has substantial mass in the low-score region; CReL therefore ranks it last. By contrast, SD3.5-L does not have the highest average DINO score, but it provides a better lower-performance profile and achieves the best CReL-DINO score. These results show that CReL captures whether high average performance is preserved near the calibrated reliability boundary.

## 5. Conclusion and Future Works

We introduce a worst-case reliability metric based on conformal calibration for evaluating conditional generative models, offering a more interpretable notion of trustworthiness than traditional single-output metrics. We further propose Conformal ReLiability (CReL), a flexible computational framework that supports most common and bespoke similarity measures. Future work will extend CReL to more complex settings such as video generation, 3D reconstruction, and embodied robotics, where one-to-many, many-to-one, or many-to-many mappings call for new joint latent representations and calibration strategies to maintain robust guarantees.

## Impact Statement

This paper presents work whose goal is to advance the field of Machine Learning. There are many potential societal consequences of our work, none which we feel must be specifically highlighted here.

## Acknowledgements

This work was supported by the State Key Program of National Natural Science Foundation of China under Grant No. 12331009, and the Young Scientists Fund of the National Natural Science Foundation of China under Grant No. KRH2305058.

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

# A. Algorithm Details

In this section, we introduce our implementation of the VAE and the Stable Diffusion model.

## A.1. Variational Auto Encoder

The variational lower bound of the model is written as follows (Sohn et al., 2015):

$$\log p_\theta(\mathbf{Y}) \geq \underbrace{\mathbb{E}_{q_\phi(\mathbf{Z}|\mathbf{Y})}\left[\log p_\theta(\mathbf{Y}|\mathbf{Z})\right] - D_{\mathrm{KL}}\left(q_\phi(\mathbf{Z}|\mathbf{Y})\|p(\mathbf{Z})\right)}_{\mathcal{L}_{\mathrm{ELBO}}}, \tag{9}$$

where $\mathcal{L}_{\mathrm{ELBO}}$ denotes the ELBO objective to be maximized. Here:

- $\mathbf{Y}$: response variables (target)

- $\mathbf{Z}$: latent variables

- $q_\phi(\mathbf{Z}|\mathbf{Y})$: inference model (encoder)

- $p_\theta(\mathbf{Y}|\mathbf{Z})$: generative model (decoder)

The latent prior is fixed as $p(\mathbf{Z}) = \mathcal{N}(\mathbf{0}, \mathbf{I})$. The decoder outputs are denoted as $\widehat{\mathbf{Y}}$.

To incorporate task-specific metric $\rho(\mathbf{Y}, \widehat{\mathbf{Y}})$ (where higher values indicate better performance), we reformulate the likelihood as an energy-based model:

$$p_\theta(\mathbf{Y}|\mathbf{Z}) \propto \exp\left(\lambda \cdot \rho(\mathbf{Y}, \widehat{\mathbf{Y}})\right), \tag{10}$$

where $\lambda > 0$ is a temperature parameter. The intractable normalization constant:

$$C(\mathbf{Z}) = \int \exp\left(\lambda \cdot \rho(\mathbf{Y}, \widehat{\mathbf{Y}})\right) d\mathbf{Y} \tag{11}$$

is omitted during optimization following energy-based modeling conventions (LeCun et al., 2006).

Substituting into the ELBO definition gives:

$$\mathcal{L}_{\mathrm{ELBO}} = \mathbb{E}_{q_\phi(\mathbf{Z}|\mathbf{X},\mathbf{Y})}\left[\lambda \cdot \rho(\mathbf{Y}, \widehat{\mathbf{Y}}) - \log C(\mathbf{Z})\right] - D_{\mathrm{KL}}\left(q_\phi(\mathbf{Z}|\mathbf{Y})\|p(\mathbf{Z})\right). \tag{12}$$

Approximating $\log C(\mathbf{X}, \mathbf{Z})$ as constant for gradient-based optimization yields:

$$\mathcal{L}_{\mathrm{ELBO}} \approx \mathbb{E}_{q_\phi(\mathbf{Z}|\mathbf{X},\mathbf{Y})}\left[\lambda \cdot \rho(\mathbf{Y}, \widehat{\mathbf{Y}})\right] - D_{\mathrm{KL}}\left(q_\phi(\mathbf{Z}|\mathbf{Y})\|p(\mathbf{Z})\right). \tag{13}$$

The final loss function $\mathcal{L} = -\mathcal{L}_{\mathrm{ELBO}}$ is composed of two parts:

$$\mathcal{L} = \lambda \cdot \mathcal{L}_\rho + \beta \cdot \mathcal{L}_{\mathrm{KL}}, \tag{14}$$

where $\mathcal{L}_\rho = -\mathbb{E}_{q_\phi(\mathbf{Z}|\mathbf{X},\mathbf{Y})}\left[\rho(\mathbf{Y}, \widehat{\mathbf{Y}})\right]$ is the metric-driven reconstruction term with $\lambda$ setting to 1, and $\mathcal{L}_{\mathrm{KL}} = D_{\mathrm{KL}}\left(q_\phi(\mathbf{Z}|\mathbf{Y})\|p(\mathbf{Z})\right)$ is the KL regularization term with $\beta$ controlling its strength.

**VAE vs. CVAE.** In (Feldman et al., 2023), the authors used the conditional variational auto-encoder (Sohn et al., 2015), where the inference model $q_\phi(\mathbf{Z}|\mathbf{Y})$ and the generative model $p_\theta(\mathbf{Y}|\mathbf{Z})$ are respectively replaced with $q_\phi(\mathbf{Z}|\mathbf{X}, \mathbf{Y})$ and $p_\theta(\mathbf{Y}|\mathbf{X}, \mathbf{Z})$. However, the CVAE can be very sensitive to the input condition $\mathbf{X}$, making it easy to collapse when conditioning on a fixed $x$. Therefore, we turn to use VAE, which can well reconstruct the output even when conditioning on a fixed $x$.

To illustrate, we train both a CVAE and a VAE on the dataset $\{X_i, Y_i\}$ in the nonlinear setting. At test time, we fix $X = x$ and generate samples $\{Y_1^x, \ldots, Y_N^x\}$ from the conditional model $Y \mid X = x$. We then visualize the reconstruction regions: for the CVAE, $\{\mathcal{D}ec_{\mathrm{CVAE}}(\mathcal{E}_{\mathrm{CVAE}}(Y_i^x, x))\}_{i \leq N}$, and for the VAE, $\{\mathcal{D}ec_{\mathrm{VAE}}(\mathcal{E}_{\mathrm{VAE}}(Y_i^x, x))\}_{i \leq N}$. As shown in Fig. 5, the VAE can reconstruct outputs faithfully when conditioned on $x$, whereas the CVAE reconstructions collapse into a much smaller region.

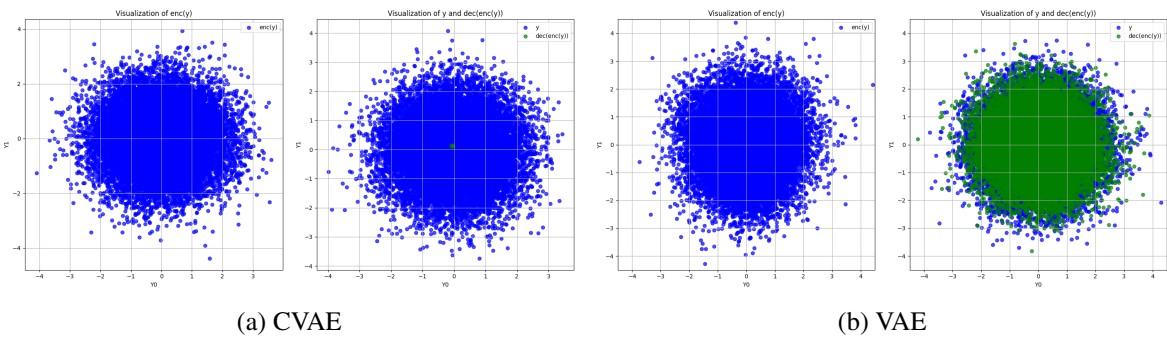

(a) CVAE              (b) VAE

*Figure 5.* The reconstruction region of CVAE (subfigure (a)) and the VAE (subfigure (b)) given a fixed $x$. In each subfigure, the left image visualizes the encoded region $\{\mathcal{E}(Y_i^x, x))\}_{i \leq N}$ in the $Z$'s space; the right image visualizes the decoded region $\{\mathcal{D}ec(\mathcal{E}(Y_i^x, x))\}_{i \leq N}$ in the $Y$'s space.

## A.2. Stable Diffusion Model

We begin by training the VAE and use its encoder $\mathcal{E}(Y_i)$ to obtain the low-dimensional latent representation $Z_i^0 = \mathcal{E}(Y_i)$ for each $i$. We then apply the diffusion process in the latent space to evolve $Z_i^0$ into $Z_i^t$, and finally reconstruct $\widehat{Y}_i$ through the decoder $\mathcal{D}ec_\phi(Z_i^t)$.

First, we consider the forward noising process:

$$q(z_t \mid z_{t-1}) = \mathcal{N}\left(z_t \,\Big|\, \sqrt{1 - \beta_t}\, z_{t-1}, \, \beta_t I\right),$$

which admits the closed-form

$$q(z_t \mid z_0) = \mathcal{N}(z_t \mid \sqrt{\alpha_t}\, z_0, \, (1 - \alpha_t)I), \quad \alpha_t = \prod_{s=1}^{t}(1 - \beta_s).$$

Thus, at step $T$, we have $z_T \sim \mathcal{N}(0, I)$. The reverse process is parameterized by a neural network $\epsilon_\theta$ (where we choose MLP on synthetic data), which predicts the noise component of $z_t$. Conditioning on $x$, the model learns to approximate

$$\epsilon_\theta(z_t, t, x) \approx \epsilon, \; \epsilon \sim \mathcal{N}(0, I).$$

The denoising distribution is then given by:

$$p_\theta(z_{t-1}|z_t, x) = \mathcal{N}(\mu_\theta(z_t, t, x), \Sigma_t I),$$

with the mean parameter

$$\mu_\theta(z_t, t, x) = \frac{1}{\sqrt{1 - \beta_t}}\left(z_t - \beta_t\, \epsilon_\theta(z_t, t, x)\right).$$

The training is based on denoising score matching. Given $(x, y)$ sampled from the dataset, we encode $z_0 = \mathcal{E}_\phi(y)$, draw $t \sim \text{Unif}(\{1, \ldots, T\})$, and generate $z_t$ via the forward process. The objective is

$$\mathcal{L}(\theta) = \mathbb{E}_{z_0, x, t, \epsilon}\left[\|\epsilon - \epsilon_\theta(z_t, t, x)\|_2^2\right].$$

At inference, one begins with Gaussian noise $z_T \sim \mathcal{N}(0, I)$ and applies the learned reverse process to obtain a latent $z_0$. The decoder then maps $z_0$ back to image space:

$$\widehat{Y} = \mathcal{D}ec_\phi(z_0).$$

## A.3. Computing Score

To compute the score $E_i^+$ in (6), we view this as a quadratic programming problem bounded by multiple inequalities:

$$\min_a \|a - Z_i\|^2 \text{ s.t.} \mathbf{u}_k^T a \geq f_\beta(X_i, \mathbf{u}_k), \|\mathbf{u}_k\| = 1, \forall k = 1, ..., K$$

Now our goal is to find a point $a^\star$ that minimizes the distance to the target point $Z_i$ and satisfies all the half-space constraints. The score takes the value of the Euclidean distance from $Z_i$ to $a^\star$.

## A.4. Proofs of Section 3

*Proof of Proposition 3.5.* We now show that for any $\gamma$, $S^\gamma(x)$ is convex and compact. For any $z_1, z_2 \in S^{\gamma_{\text{cal}}}(x)$, we have $a_1 \in R_{\mathcal{Z}}(x), a_2 \in R_{\mathcal{Z}}(x)$ respectively such that $d(a_1, z_1) \leq \gamma$ and $d(a_2, z_2) \leq \gamma$. Besides, as $R_{\mathcal{Z}}(x)$ is convex, we have $\alpha a_1 + (1-\alpha)a_2 \in R_{\mathcal{Z}}(x)$. Then for any $0 < \alpha < 1$, we have:

$$
\min_{a \in R_{\mathcal{Z}}(x)} d(a, \alpha z_1 + (1-\alpha)z_2) = d(\alpha a + (1-\alpha)a, \alpha z_1 + (1-\alpha)z_2)
$$
$$
\leq d(\alpha a_1 + (1-\alpha)a_2, \alpha z_1 + (1-\alpha)z_2)
$$
$$
\overset{(1)}{\leq} \alpha\|a_1 - z_1\|_2 + (1-\alpha)\|a_2 - z_2\|_2 \leq \gamma,
$$

where "(1)" is due to the jointly convexity of $d(\cdot, \cdot)$. The compactness follows from the compactness of $R_{\mathcal{Z}}(X)$ and that $\gamma_{\text{cal}}$ is bounded. $\qquad\square$

# B. Experimental Details

## B.1. Synthetic Data Details

**Linear data generation.** The generation of the condition vector $X$ and the response variable $Y$ in the *linear* version of the synthetic data is defined as follows:

$$
\begin{aligned}
X &\sim \text{Uniform}(0.8, 3.2)^p, \\
A &\sim \mathcal{N}(0,1)^{p \times d}, \\
\epsilon &\sim \mathcal{N}(0, \sigma^2)^d, \\
Y &= XA + \epsilon,
\end{aligned}
\tag{15}
$$

where $\text{Uniform}(a,b)$ is a uniform distribution on the interval $(a,b)$, $X \in \mathbb{R}^p$ is the condition vector, $A \in \mathbb{R}^{p \times d}$ is the coefficient matrix, $\epsilon \in \mathbb{R}^d$ is Gaussian noise with variance $\sigma^2$, and $Y \in \mathbb{R}^d$ is the response variable.

**Nonlinear data generation.** The *nonlinear* version of the synthetic data is generated as follows:

$$
\begin{aligned}
X &\sim \text{Uniform}(0.8, 3.2)^p, \\
A &\sim \mathcal{N}(0,1)^{p \times d}, \\
B &\sim \mathcal{N}(0,1)^{p \times d}, \\
\epsilon &\sim \mathcal{N}(0, \sigma^2)^d, \\
Y &= XA + X^2 B + \epsilon,
\end{aligned}
\tag{16}
$$

where $X \in \mathbb{R}^p$, $A \in \mathbb{R}^{p \times d}$, $B \in \mathbb{R}^{p \times d}$, $\epsilon \in \mathbb{R}^d$, and $Y \in \mathbb{R}^d$. The term $X^2$ denotes element-wise squaring of $X$.

## B.2. Implementation Details

**Network architectures.**

- *VAE Encoder/Decoder Hidden Dimensions:*
    - Synthetic data: $[64, 128, 256, 256, 128, 64]$
    - Image-to-text task: $[128, 256, 512, 512, 256, 128]$

- *Stable Diffusion Denoiser:*
    - MLP hidden dimensions: $[128, 256, 128]$
    - Time embedding dimension: $128$

- *DQR Network:*
    - Hidden dimensions: $[8, 16, 8]$

- Dropout (rate 0.1) and batch normalization are applied for the image-to-text dataset.

**Training hyperparameters.**

- *VAE:*
  - Learning rate: $1 \times 10^{-3}$
  - Activation: leaky ReLU (slope 0.2)

- *Stable Diffusion:*
  - Learning rate: $1 \times 10^{-4}$
  - Number of diffusion timesteps (training): 1000

**DQR directions.**

- Each gradient step uses 1024 distinct directions, sampled from a fixed set of 2048 directions generated before training.

**Discretization.**

- Number of grid points to decode region in $\mathcal{Z}$ space: $2 \times 10^4$

- Feldman grid in $\mathcal{Z}$ space: $2 \times 10^4$

- Feldman grid in $\mathcal{Y}$ space: $2 \times 10^4$

- Number of grid points for area calculation in $\mathcal{Y}$ space: $2 \times 10^4$

**Hardware.**

- Synthetic data simulations: NVIDIA RTX A6000 GPU (48GB VRAM)

- Image-to-text task: NVIDIA H100 GPU (80GB HBM3)

## C. Ablation Study

### C.1. Effect of the KL Regularization Weight in VAE

To investigate the effect of the KL regularization weight ($\beta$) in the VAE training loss, we conduct an ablation study on the *nonlinear* synthetic data, following the same setup as Section 4.1. As shown in Table 4, when $\beta = 0.001$, our method achieves both the target coverage ($\alpha = 0.1$) and a compact informative region. Therefore, we set $\beta = 0.001$ for all experiments.

*Table 4.* Ablation study on the effect of the KL regularization weight $\beta$ in the VAE loss. The table reports the coverage ratios and the area of the region $C_{\mathcal{Y}}$ on the *nonlinear* synthetic dataset for different values of $\beta$. The target nominal level is $\alpha = 0.1$.

| Metric | $\beta = 0.1$ | $\beta = 0.01$ | $\beta = 0.001$ | $\beta = 0.0001$ | $\beta = 0.00001$ |
|---|---|---|---|---|---|
| coverage of $R_{\mathcal{Z}}$ | 0.5570 | 0.5525 | 0.4465 | 0.4203 | 0.4910 |
| coverage of $C_{\mathcal{Z}}$ | 0.8883 | 0.8995 | 0.8953 | 0.8908 | 0.8945 |
| coverage of $R_{\mathcal{Y}}$ | 0.9200 | 0.7280 | 0.5387 | 0.5060 | 0.5730 |
| coverage of $C_{\mathcal{Y}}$ | 0.9945 | 0.9525 | 0.8915 | 0.8832 | 0.8895 |
| area of $C_{\mathcal{Y}}$ | 1044.54 | 320.58 | 232.73 | 213.26 | 249.41 |

*Table 5.* Ablation study of the SD conditional denoiser: coverage and area for different inference steps $T$ on the *nonlinear* synthetic dataset ($\alpha = 0.1$).

| Metric | $T = 10$ | $T = 20$ | $T = 30$ | $T = 40$ | $T = 50$ |
|:---:|:---:|:---:|:---:|:---:|:---:|
| coverage of $R_{\mathcal{Z}}$ | 0.4452 | 0.4725 | 0.4412 | 0.5503 | 0.5570 |
| coverage of $C_{\mathcal{Z}}$ | 0.8968 | 0.9025 | 0.8952 | 0.9000 | 0.8998 |
| coverage of $R_{\mathcal{Y}}$ | 0.5595 | 0.6218 | 0.6338 | 0.8055 | 0.8475 |
| coverage of $C_{\mathcal{Y}}$ | 0.9065 | 0.9405 | 0.9675 | 0.9773 | 0.9883 |
| area of $C_{\mathcal{Y}}$ | 239.05 | 285.93 | 367.64 | 405.89 | 525.12 |

*Table 6.* Ablation study of the SD unconditional denoiser: coverage and area for different inference steps $T$ on the *nonlinear* synthetic dataset ($\alpha = 0.1$).

| Metric | $T = 10$ | $T = 20$ | $T = 30$ | $T = 40$ | $T = 50$ |
|:---:|:---:|:---:|:---:|:---:|:---:|
| coverage of $R_{\mathcal{Z}}$ | 0.4460 | 0.5320 | 0.5333 | 0.5507 | 0.5620 |
| coverage of $C_{\mathcal{Z}}$ | 0.9008 | 0.8900 | 0.8988 | 0.8960 | 0.9058 |
| coverage of $R_{\mathcal{Y}}$ | 0.5668 | 0.6893 | 0.7418 | 0.8085 | 0.8613 |
| coverage of $C_{\mathcal{Y}}$ | 0.9203 | 0.9468 | 0.9638 | 0.9790 | 0.9923 |
| area of $C_{\mathcal{Y}}$ | 271.26 | 309.56 | 370.03 | 415.60 | 542.00 |

## C.2. Choice of Latent Generative Model

To explore the effect of different latent generative models in our framework, we compare the Variational Autoencoder (VAE) and Stable Diffusion (SD) models on the *nonlinear* synthetic dataset, using the same experimental setup as Section 4.1. For the VAE, the KL regularization weight is set to $\beta = 0.001$. For the SD model, we use an MLP network as the denoiser; for implementation details regarding SD, please refer to Section B.2.

**Ablation study: SD denoiser architecture and inference steps.**  We first conduct an ablation study on the SD model to investigate the effect of (a) whether the denoiser is conditioned on input, and (b) the number of inference steps $T$ (ranging from 10 to 50). We set the target nominal level to $\alpha = 0.1$. As shown in Tables 5 and 6, all values of $T$ achieve the target coverage, with $T = 10$ providing the tightest coverage and, consequently, the least conservative region Therefore, we use the conditional denoiser with $T = 10$ in all subsequent SD experiments.

**Comparison between SD and VAE as latent generative models.**  Finally, we compare the performance of SD (with conditional denoiser, $T = 10$) and VAE as the latent generative model in our CReL framework, evaluating both the coverage and the area on the *nonlinear* synthetic dataset for two nominal levels ($\alpha = 0.02, 0.10$), as summarized in Table 7. Both models achieve the target coverage, but the VAE consistently produces a more compact (informative) covered region in $\mathcal{Y}$. Based on these results, we use the VAE as the default latent generative model in all main experiments, due to its greater informativeness while maintaining desired coverage.

## C.3. Robustness of Coverage Guarantees to Similarity Metric

CReL is designed to be metric-agnostic, successfully quantifying reliability with respect to a user's specific concern. To demonstrate that our coverage guarantees are independent of the metric choice, we conducted a new simulation experiment

*Table 7.* Comparison of VAE and SD as latent generative models in the CReL framework on the *nonlinear* synthetic dataset. Coverage and area metrics are reported for different $\alpha$.

| $\alpha$ | Coverage | | | | Area in $\mathcal{Y}$ | |
|:---:|:---:|:---:|:---:|:---:|:---:|:---:|
| | VAE-$\mathcal{Z}$ | VAE-$\mathcal{Y}$ | SD-$\mathcal{Z}$ | SD-$\mathcal{Y}$ | VAE | SD |
| 0.02 | 0.9770 | 0.9760 | 0.9810 | 0.9843 | 398.51 | 432.99 |
| 0.10 | 0.8953 | 0.8915 | 0.8968 | 0.9065 | 232.73 | 239.05 |

using the Mean Absolute Error (MAE) as the metric $\rho$. As shown in Tab. 8, the empirical coverage remains valid and closely tracks the nominal levels, just as it did for MSE in the main text.

*Table 8.* Coverage ratios and areas on the nonlinear synthetic dataset using MAE as the similarity metric.

| $\alpha$ | Coverage in $\mathcal{Z}$ | Coverage in $\mathcal{Y}$ | Area in $\mathcal{Y}$ |
|---|---|---|---|
| 0.02 | 0.9743 | 0.9735 | 407.0 |
| 0.04 | 0.9563 | 0.9540 | 353.8 |
| 0.06 | 0.9403 | 0.9405 | 295.5 |
| 0.08 | 0.9100 | 0.9147 | 267.0 |
| 0.10 | 0.8905 | 0.8938 | 243.1 |

## C.4. Comparison with PCP

We compare CReL with PCP (Wang et al., 2022b) on the nonlinear simulation dataset at $\alpha = 0.1$. CReL uses the VAE latent generative model. PCP uses its default multidimensional setting with $K = 1000$, `caltype=uniform`, and `md_type=kmn`; PCP results are averaged over 5 independent runs.

*Table 9.* Comparison with PCP on the nonlinear simulation dataset at $\alpha = 0.1$.

| Method | Coverage in $\mathcal{Y}$ | Area in $\mathcal{Y}$ |
|---|---|---|
| CReL (VAE) | 0.8915 | 232.70 |
| PCP | 0.9001 | 854.24 |

## C.5. Sensitivity to LGM Reconstruction Error

We conduct a sensitivity analysis on the nonlinear simulation dataset with the VAE latent generative model and $\alpha = 0.1$. During VAE training, we monitor epochs $[1, 5, 10, 100, 183, 300]$, where epoch 183 is the best epoch. Table 10 reports representative checkpoints with available coverage and area results. Coverage is stable once the reconstruction loss becomes sufficiently small, but fails when the reconstruction error is large.

*Table 10.* Sensitivity analysis of coverage with respect to VAE reconstruction error on the nonlinear simulation dataset at $\alpha = 0.1$.

| Epoch | 1 | 5 | 10 | 100 | 183 | 300 |
|---|---|---|---|---|---|---|
| $\mathcal{L}$ | 0.3740 | 0.0197 | 0.0112 | 0.0086 | 0.0080 | 0.0084 |
| Coverage in $\mathcal{Y}$ | 0.0205 | 0.8468 | 0.8830 | 0.8972 | 0.8915 | 0.8955 |
| Area in $\mathcal{Y}$ | 23.24 | 287.12 | 278.39 | 235.49 | 232.70 | 246.78 |

The failure at early epochs is caused by large reconstruction error, while the coverage becomes close to the target level after epoch 10. This supports that a sufficiently trained LGM can avoid substantial bias in the final reliability evaluation.

## C.6. Computational Cost of LGM Training

We report the empirical overhead of training the VAE used as the latent generative model in Table 11. Experiments are run on an NVIDIA GeForce RTX 4090 48GB GPU with batch size 512. This cost is incurred only once for each target generative model as an offline preprocessing step, after which the calibrated latent-space pipeline can be reused to evaluate many test examples.

## C.7. Effect of Latent Space Dimensionality in VAE

To determine the appropriate dimensionality of the VAE latent space for the image-to-text task, we conduct an ablation study by varying the latent dimension $r$ and evaluating its impact on the loss value (BERT-SIM, BLIP-large). The results, shown in Table 12, indicate that the loss remains relatively stable across a wide range of latent dimensions. Based on these observations, we empirically set $r = 50$ for all image-to-text experiments.

*Table 11.* Training cost of the VAE latent generative model.

| Setting | Metric | Epochs | Time (min) | Peak VRAM (GB) |
|---|---|---|---|---|
| Synthetic | MSE | 385 | 2.80 | 0.06 |
| Image-to-text (BLIP-base) | CLIP-SIM | 484 | 15.11 | 0.22 |

*Table 12.* Ablation study on the dimensionality of the VAE latent space in the image-to-text task. The table reports the loss value $\mathcal{L}$ for different choices of the latent dimension $r$.

| $r$ | 10 | 20 | 50 | 100 | 200 | 300 |
|---|---|---|---|---|---|---|
| $\mathcal{L}$ | 0.0442 | 0.0418 | 0.0418 | 0.0442 | 0.0422 | 0.0422 |

## C.8. Effect of Initial Points

We analyze how the number of initial points ($z_0$) affects the accuracy of reliability estimates. As described in Section 3, the optimization of 8 combines the projected gradient descent and random search, where $\text{num}_{z_0}$ is a key hyperparameter. A larger $\text{num}_{z_0}$ improves estimation accuracy but increases computational cost.

To study its impact, we evaluate four generative models on the image-to-text task using CReL-CLIP and CReL-BERT at $\alpha = 0.1$, and four text-to-image models using CReL-CLIP and CReL-DINO at $\alpha = 0.1$. As shown in Fig. 6 and Fig. 7, increasing $\text{num}_{z_0}$ generally decreases the estimated reliability score until the curves stabilize. To balance reliability and computational cost, we set $\text{num}_{z_0} = 50$ for CReL-CLIP and CReL-BERT, and $\text{num}_{z_0} = 100$ for CReL-DINO in the text-to-image experiments.

**Initialization seed sensitivity.** We further test the sensitivity of the projected gradient descent procedure to random initialization, with results summarized in Table 13. On the image-to-text task, we evaluate BLIP-base using CLIP-SIM with $\alpha = 0.1$ and $\text{num}_{z_0} = 50$ over 10 repeated runs. The standard deviation of the resulting CReL-CLIP score is 0.00027, indicating that the optimization is highly stable across random initializations.

*Table 13.* Sensitivity of CReL-CLIP to random initialization on image-to-text evaluation.

| Task | Model | Metric | $\alpha$ | $\text{num}_{z_0}$ | Runs | Std. |
|---|---|---|---|---|---|---|
| Image-to-text | BLIP-base | CLIP-SIM | 0.1 | 50 | 10 | 0.00027 |

## D. Computational Complexity Comparison

We compare the computational complexity of our calibration scheme with the grid-based discretization method of Feldman *et al.* (Feldman et al., 2023). While their approach incurs exponential costs in both the latent and original data spaces, our method operates directly in the lower-dimensional embedding and leverages DQR for initialization, yielding a significant reduction in computational cost. Here, we denote $n_{\text{cal}} := |\mathcal{I}_{\text{cal}}|$ as the sample size in the calibration set.

**Feldman *et al.* (grid-based).** They discretize both the $r$-dimensional latent space and the $d$-dimensional original space using a uniform grid of size $m$ per axis:

- Latent space discretization: $O(m^r)$.

- Original space discretization: $O(m^d)$.

- Quantile initialization: computing the 90th percentile of all pairwise distances to obtain $\gamma_{\text{init}}$ requires $O(n_{\text{cal}} \cdot m^{2d} \cdot d \cdot \log m)$.

**Our calibration scheme.** Our method avoids costly discretization and initialization by:

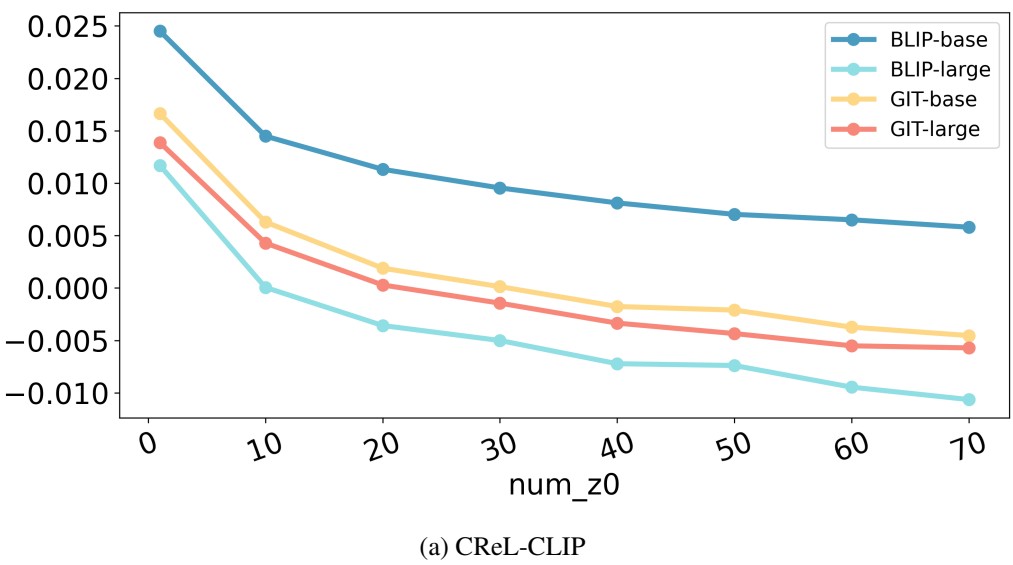

(a) CReL-CLIP

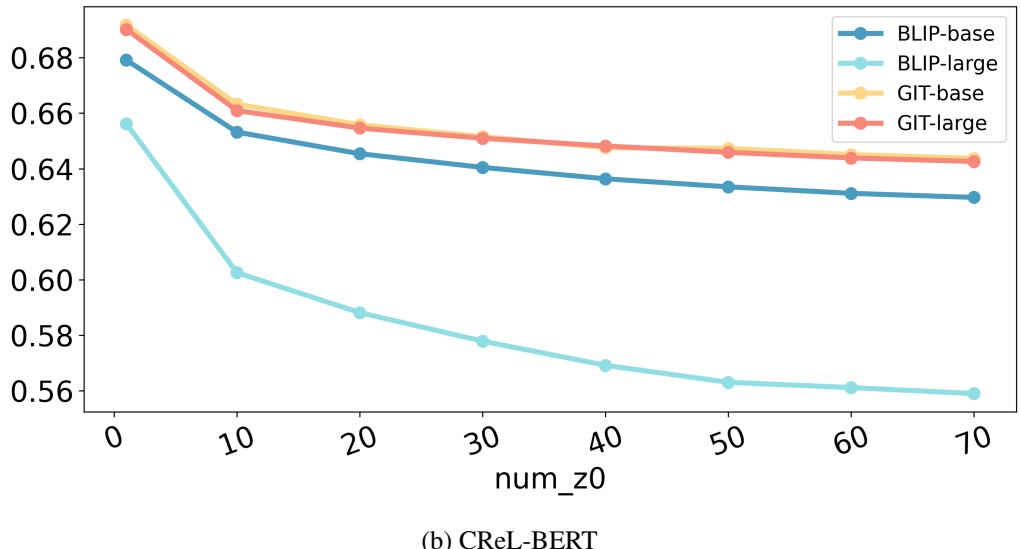

(b) CReL-BERT

*Figure 6.* Effect of the number of initial points ($z_0$) on CReL-$\rho$ for the image-to-text task.

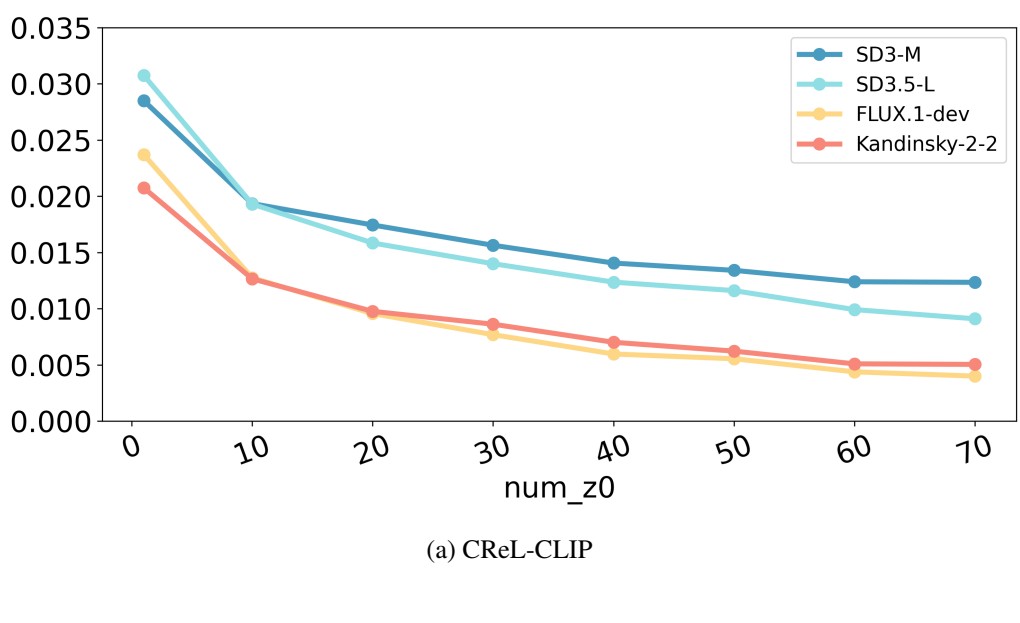

(a) CReL-CLIP

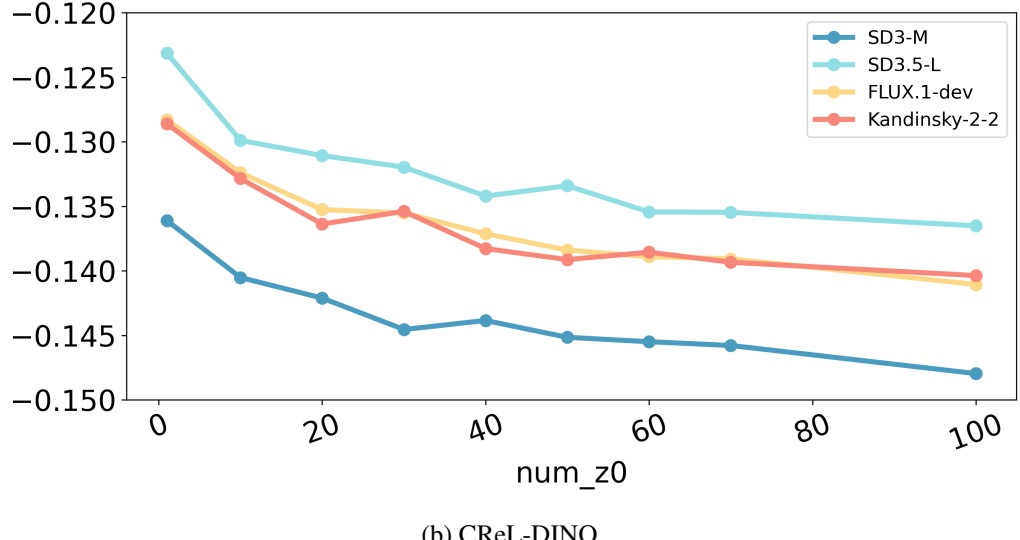

(b) CReL-DINO

*Figure 7.* Effect of the number of initial points ($z_0$) on CReL-CLIP and CReL-DINO for the text-to-image task.

- Latent-space region: directly constructing the quantile region in the $r$-dimensional embedding space, bypassing any $m$-grid.

- DQR initialization: using the region obtained from DQR as the calibration starting point, eliminating the $O(n_{\text{cal}} \cdot m^{2d} \cdot d \cdot \log m)$ step.

- Score computation: performing quadratic-programming updates in $O(n_{\text{cal}} \cdot T \cdot q \cdot r)$, where $T$ is the number of iterations and $q$ is the number of calibration directions.

**Overall Improvement.** By operating in the lower-dimensional latent space ($r \ll d$) and leveraging our calibration scheme, we reduce the computational complexity of the calibration step from $O(n_{\text{cal}} \cdot m^{2d} \cdot d \cdot \log m)$ to $O(n_{\text{cal}} \cdot T \cdot q \cdot r)$.

For empirical runtime comparisons, see Appendix E.

## E. Calibration Scheme Runtime Comparison

### E.1. Setup

**Simulations on synthetic data.** We evaluate the runtime of our calibration scheme against the grid-based discretization method of Feldman *et al.* (Feldman et al., 2023) using a *linear* synthetic dataset (generation details in Appendix B.1). We generate $n = 50{,}000$ samples $p = 50$ and $d = 20$, and split them as follows:

| | | |
|---|---|---|
| VAE training set: | 60% | (30,000 samples) |
| DQR training set: | 24% | (12,000 samples) |
| Calibration set: | 8% | (4,000 samples) |
| Test set: | 8% | (4,000 samples) |

The calibration set is used to measure the runtime of both schemes.

**Implementation details.** For the VAE, we vary the latent dimension $r$ from 2 to 12 and fix the loss weight $\beta = 0.01$. In the grid-based scheme (Feldman et al., 2023), we use a uniform grid of size $m = 5$ per latent axis and fix the total number of grid points in the original space to 300,000 to control memory usage. For DQR, the input size is $p + d$, and each gradient step uses 1024 directions with $\alpha = 0.1$. All data undergo $L_2$ normalization before training. Further details are provided in Appendix B.2.

### E.2. Results

We report the total calibration runtime (in seconds) for both our calibration scheme and the grid-based discretization method of Feldman *et al.* across different latent dimensions. Several key observations emerge:

Our method exhibits near-linear growth in runtime with respect to $r$. Starting at approximately 16.4 s for $r = 2$, the total runtime increases modestly to about 63.5 s at $r = 12$, corresponding to an average incremental cost of under 5 s per additional latent dimension. This behavior is consistent with our theoretical complexity (Appendix D), in which $r$ enters only linearly.

In contrast, the grid-based approach exhibits exponential growth: its calibration time increases from 116.6 s at $r = 2$ to 433.3 s at $r = 8$ and finally to 122,795.0 s at $r = 12$. Correspondingly, our method is over $8\times$ faster at $r = 8$ (51.37 s vs. 433.3 s) and nearly $1{,}930\times$ faster at $r = 12$ (63.54 s vs. 122,795.0 s), rendering the grid-based scheme infeasible for moderate-to-high latent dimensions.

These empirical results corroborate our theoretical complexity reduction and demonstrate that by operating directly in the lower-dimensional embedding space, our calibration scheme remains computationally feasible even as $r$ grows large. This efficiency gain is critical for scaling conformal calibration to high-dimensional prediction tasks.

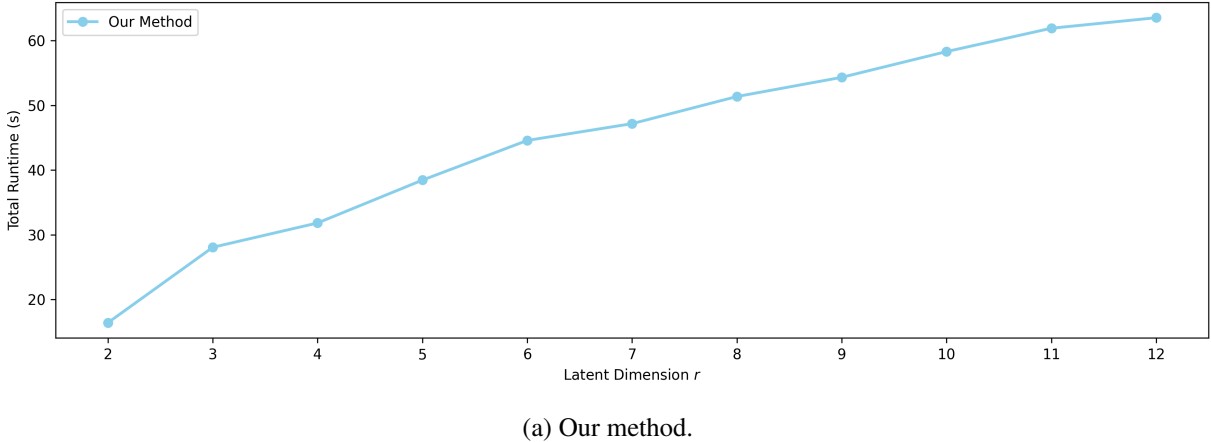

(a) Our method.

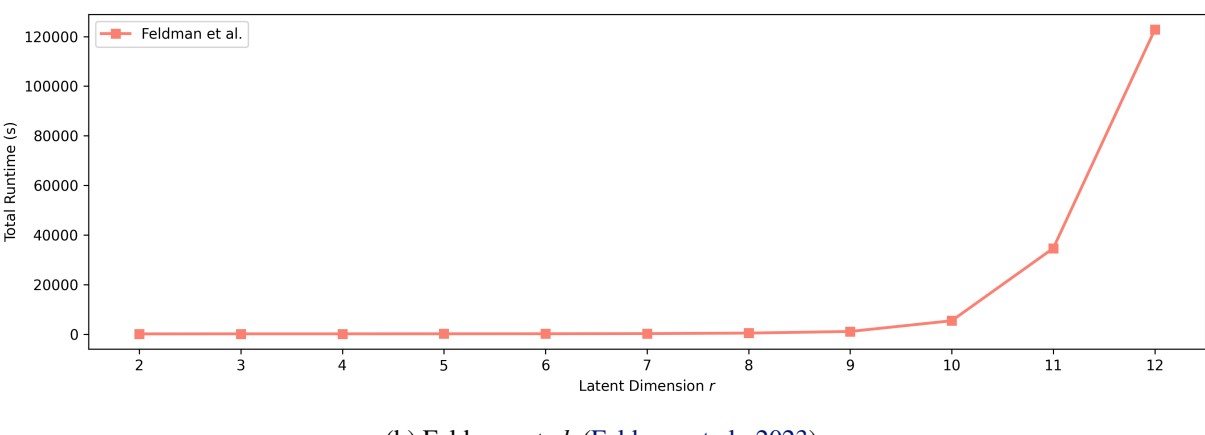

(b) Feldman *et al.* (Feldman et al., 2023).

*Figure 8.* Comparison of total calibration runtime (in seconds) across different latent dimensions $r$. Our method exhibits favorable scalability, whereas the grid-based approach of (Feldman et al., 2023) incurs significantly higher computational cost as $r$ increases.

## F. Additional Results

### F.1. Qualitative Results of Image-to-Text Task

We provide additional qualitative examples for the image-to-text task. Figure 9 and Figure 10 present the results of CReL-CLIP and CReL-BERT under $\alpha = 0.1$, respectively.

### F.2. Experiments on Text-to-Image Task

**Dataset and preprocessing.** We use the MS-COCO 2014 validation set (Lin et al., 2014) ($40,504$ image-caption pairs), and split it into $75\%$ for VAE training, $15\%$ for DQR, $5\%$ for calibration, and $5\%$ for testing. We evaluate four models: SD3-M (Esser et al., 2024), SD3.5-L (Esser et al., 2024), FLUX.1-dev (Labs, 2024), and Kandinsky-2.2 (Shakhmatov et al., 2023), all at image size $512 \times 512$. We use two metrics. CLIP-SIM is the cosine similarity between the CLIP text feature of the caption and the CLIP image feature of the generated image. DINO-SIM is the cosine similarity between DINOv2-base image features of the generated and ground-truth images (Oquab et al., 2023).

**Implementation details.** For CLIP-SIM, image and text features are extracted using CLIP ViT-L/14 (Radford et al., 2021), and both feature types have dimension $p = d = 768$. For DINO-SIM, generated and ground-truth image features are extracted using DINOv2-base (Oquab et al., 2023), whose feature dimension is also 768. All generation models use 50 inference steps and a guidance scale of 7.0. For VAE, we set $r = 50$ and use $\beta = 0.001$ for KL regularization. For DQR,

the input size is $p + d$, and each gradient step uses $1,024$ directions with $\alpha = 0.1$. All data are $L_2$-normalized before training. During optimization, we initialize the procedure with $50$ starting points for CLIP-SIM and $100$ starting points for DINO-SIM. More details can be found in Appendix C.8.

**Quantitative comparison.** We compare four text-to-image models under CLIP-SIM and DINO-SIM. Tab. 3 reports the average scores and the corresponding CReL scores, while Fig. 11 visualizes the score distributions.

For CLIP-SIM, Fig. 11a shows that all four models have highly overlapping distributions. The box plots and violin shapes are close, especially for SD3-M, SD3.5-L, and Kandinsky-2.2. This means that Kandinsky-2.2's highest average CLIP score in Tab. 3 should not be interpreted as clear dominance over the other models. Rather, it reflects a small advantage in the central or upper part of the CLIP distribution. CReL changes the comparison because it asks whether this advantage is preserved near the lower-performance boundary. Under this view, SD3-M obtains the highest CReL-CLIP score. Although SD3-M does not have the highest average CLIP score, its distribution remains competitive in the central region and does not rely on a clearly separated high-score tail. Thus, its semantic alignment is more stable at the calibrated reliability boundary. Kandinsky-2.2 drops from first under average CLIP to third under CReL-CLIP because its small average advantage does not translate into a better lower-bound reliability score. FLUX.1-dev stays last because its CLIP distribution is slightly shifted downward and shows a more pronounced low-score side.

For DINO-SIM, the pattern is more pronounced. Fig. 11b shows much broader distributions than CLIP-SIM, indicating that image-level structural similarity varies substantially across samples. In this setting, the average score can be strongly influenced by the upper part of the distribution. SD3-M is the clearest example: it has many high-DINO samples and therefore achieves the highest average DINO score in Tab. 3. However, its violin plot also shows a wide spread and substantial low-score mass. CReL penalizes this weak lower-performance region, so SD3-M drops to the lowest CReL-DINO score. SD3.5-L, in contrast, has a slightly lower average DINO score but a more favorable reliability profile near the lower-performance boundary, leading to the best CReL-DINO score. Kandinsky-2.2 and FLUX.1-dev have lower average DINO scores; among them, Kandinsky-2.2 obtains a better CReL-DINO score, consistent with its slightly more favorable lower-side behavior in the distribution.

**Qualitative results: CReL effectively identifies misalignments.** Figure 12 provides qualitative examples illustrating that our calibrated metric better reflects generation reliability compared to the uncalibrated single-sample metric. Standard metrics like CLIP often assign high scores to images that capture the general theme but miss crucial semantic details specified in the prompt. For instance, given the prompt "A small bedroom with sofa at the end of the bed", only the FLUX.1-dev model correctly generates the specific spatial arrangement, yet standard CLIP ranks it lower than the failing models; in contrast, CReL correctly identifies it as the most reliable model with the highest score. Similarly, for "A cat on a suitcase is reaching for a pillow", SD3.5-L successfully depicts the "reaching" action while others fail, but CLIP ranks it last; CReL provides a more nuanced evaluation that better aligns with this semantic fulfillment. Conversely, CReL effectively penalizes hallucinations or failures that CLIP misses: in the case of "a dog is laying on his back...", SD3-M fails to generate the correct pose but receives a high CLIP score, whereas CReL's reliability assessment reflects the risk of this semantic failure. Finally, for "A giant clock tower window is looked through by many", where SD3.5-L (CLIP rank 1) generates inconsistent content and SD3-M misses key semantics, FLUX.1-dev perfectly captures the scene and is accurately ranked first by CReL. These results demonstrate that CReL effectively detects fine-grained semantic discrepancies that standard metrics miss, quantifying model reliability without solely relying on average performance.

|  |  | CLIP | CReL-CLIP |
|---|---|---|---|
|  | **GT caption:** A cat sitting on top of a pile of books in a city. | | |
|  | **BLIP-base:** a cat sitting on a pile of books | 0.2698 | 0.0097 |
|  | **BLIP-large:** araffe cat sitting on top of a pile of books on a sidewalk | 0.2919 | -0.0026 |
|  | **GIT-base:** a cat sitting on books in a cafe | 0.2932 | -0.0105 |
|  | **GIT-large:** a cat sitting on top of a book on a table. | 0.2737 | -0.0313 |
|  | **GT caption:** a woman using a white laptop on the bed | | |
|  | **BLIP-base:** a boy laying on a bed | 0.1917 | 0.0134 |
|  | **BLIP-large:** arafed woman laying on bed using laptop computer with pink sheets | 0.2082 | 0.0385 |
|  | **GIT-base:** a woman laying on a bed using a laptop. | 0.2246 | 0.0358 |
|  | **GIT-large:** a young man laying on a bed looking at a laptop. | 0.2769 | 0.0114 |
|  | **GT caption:** The man is laying out in the sand at the beach | | |
|  | **BLIP-base:** a man laying on the beach | 0.2317 | 0.0224 |
|  | **BLIP-large:** there is a man laying on a beach with a surfboard | 0.2468 | -0.0118 |
|  | **GIT-base:** a man laying on the beach in the sand | 0.2412 | 0.0010 |
|  | **GIT-large:** a man laying on the beach with his arms stretched out. | 0.2574 | 0.0315 |
|  | **GT caption:** A horse that is in the middle of a patch of flowers. | | |
|  | **BLIP-base:** a flower garden with many different flowers | 0.2517 | 0.0226 |
|  | **BLIP-large:** arafed flower garden with a dog in the middle of it | 0.2818 | 0.0057 |
|  | **GIT-base:** a dog in a flower bed | 0.2799 | 0.0104 |
|  | **GIT-large:** a flower garden with a horse in the middle. | 0.2922 | 0.0241 |
|  | **GT caption:** a double decked bus parked by a stadium | | |
|  | **BLIP-base:** a red bus parked in front of a building | 0.2546 | 0.0017 |
|  | **BLIP-large:** arafed bus parked in front of a large tent on a hill | 0.2568 | -0.0192 |
|  | **GIT-base:** a red bus in the parking lot | 0.2250 | -0.0227 |
|  | **GIT-large:** a red double decker bus parked in a parking lot. | 0.2218 | 0.0059 |
|  | **GT caption:** A little girl sitting at the end of a bed looking at a teddy bear. | | |
|  | **BLIP-base:** a little girl sitting on a bed with a teddy bear | 0.2958 | 0.0137 |
|  | **BLIP-large:** there is a little girl sitting on a bed with a teddy bear | 0.2742 | 0.0115 |
|  | **GIT-base:** a little boy sitting on a bed with a stuffed animal. | 0.3096 | -0.0280 |
|  | **GIT-large:** a child sitting on a bed next to a teddy bear. | 0.3176 | 0.0037 |
|  | **GT caption:** A man in a suit and tie standing in the desert. | | |
|  | **BLIP-base:** a man in a suit and tie standing in a field | 0.2758 | -0.0032 |
|  | **BLIP-large:** arafed man in suit and tie standing in front of a beach | 0.2864 | 0.0313 |
|  | **GIT-base:** a man standing on a beach with a suit and tie. | 0.2743 | 0.0386 |
|  | **GIT-large:** a man in a suit and tie standing on the beach. | 0.2700 | 0.0386 |

*Figure 9.* Qualitative results of image-to-text models ($\alpha = 0.1$).

| | | BERT | CReL-BERT |
|---|---|---|---|
|  | **GT caption:** A boy holds the guitar controller from Guitar Hero. | | |
| | **BLIP-base:** a young boy holding a guitar in his living room | 0.8825 | 0.7312 |
| | **BLIP-large:** boy holding a guitar in front of a television with a plant in front of him | 0.7915 | 0.6474 |
| | **GIT-base:** a boy holding a guitar and a guitar. | 0.8984 | 0.7145 |
| | **GIT-large:** a young boy holding a guitar in front of a television. | 0.8768 | 0.7184 |
|  | **GT caption:** A man carrying two traffic lights on the side of a street. | | |
| | **BLIP-base:** a man is cleaning the street | 0.8586 | 0.6700 |
| | **BLIP-large:** there is a man that is standing on a street corner with a traffic light | 0.8252 | 0.6778 |
| | **GIT-base:** a man standing on a curb holding two traffic lights. | 0.9820 | 0.8061 |
| | **GIT-large:** a man standing on a sidewalk holding a traffic light. | 0.9816 | 0.7321 |
|  | **GT caption:** A man laying on the beach next to a surfboard. | | |
| | **BLIP-base:** a man laying on the beach | 0.9117 | 0.6488 |
| | **BLIP-large:** surfers sitting on the beach with their surfboards in front of a mural | 0.8754 | 0.6341 |
| | **GIT-base:** a man laying on the beach with a surfboard. | 0.9937 | 0.6718 |
| | **GIT-large:** a man sitting on the beach with a surfboard. | 0.9859 | 0.6412 |
|  | **GT caption:** Elephant walking through the middle of the road in front of a car. | | |
| | **BLIP-base:** an elephant walking across the road | 0.8817 | 0.6545 |
| | **BLIP-large:** elephants walking down the road with cars in the background | 0.8424 | 0.6667 |
| | **GIT-base:** a large elephant walking across a road next to a car. | 0.9447 | 0.6799 |
| | **GIT-large:** an elephant walking down a road next to a car. | 0.9232 | 0.6798 |
|  | **GT caption:** an image of a black bear in the woods | | |
| | **BLIP-base:** a bear is standing in the woods | 0.9221 | 0.6526 |
| | **BLIP-large:** araffe in the woods at night with a stick in its mouth | 0.9104 | 0.4404 |
| | **GIT-base:** a black bear in the woods with a large mouth. | 0.8775 | 0.6351 |
| | **GIT-large:** a black bear walking through a forest at night. | 0.7656 | 0.7144 |
|  | **GT caption:** An old styke suitcase being used as a decorative flower pot. | | |
| | **BLIP-base:** a wooden box with a plant inside | 0.7157 | 0.6371 |
| | **BLIP-large:** there is a small box with plants inside of it on a table | 0.7423 | 0.6113 |
| | **GIT-base:** a suitcase filled with plants on top of a wooden floor. | 0.8719 | 0.6730 |
| | **GIT-large:** a suitcase with a bunch of plants inside of it | 0.7999 | 0.6546 |
|  | **GT caption:** A woman takes a picture of a train on a track. | | |
| | **BLIP-base:** a woman standing on train tracks | 0.8111 | 0.6119 |
| | **BLIP-large:** there is a woman standing on the train tracks looking at a train | 0.8076 | 0.6548 |
| | **GIT-base:** a woman standing on a train track next to a blue train. | 0.8644 | 0.6464 |
| | **GIT-large:** a woman standing on a train track next to a tunnel. | 0.8493 | 0.6252 |

*Figure 10.* Qualitative results of image-to-text models ($\alpha = 0.1$).

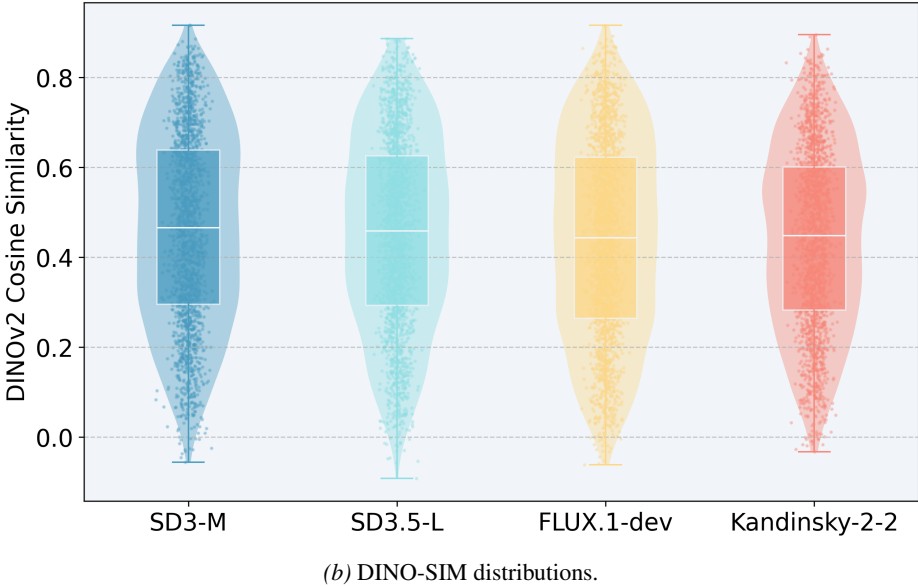

*(a)* CLIP-SIM distributions.

*(b)* DINO-SIM distributions.

*Figure 11.* Score distributions across four text-to-image models under CLIP-SIM and DINO-SIM.

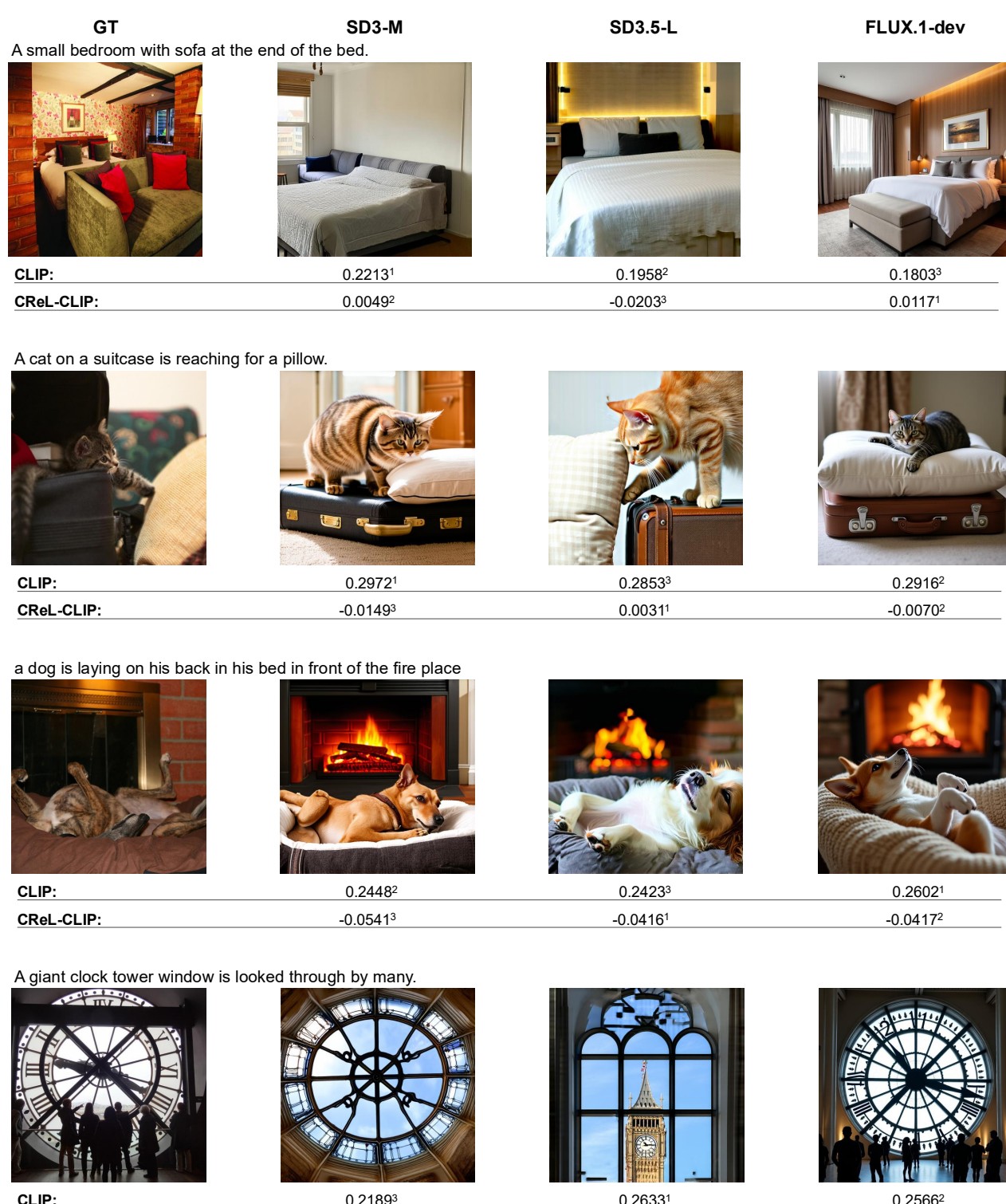

*Figure 12.* Qualitative results of text-to-image models ($\alpha = 0.1$). Superscripts denote rank.

