# OpenReview forum: "Conformal Reliability: A New Evaluation Metric for Conditional Generation"
_ICML.cc/2026/Conference — ICML 2026 regular_

### Official Review · Reviewer_w6et · 2026-03-09

**Soundness:** 3
**Presentation:** 2
**Significance:** 3
**Originality:** 3
**Overall Recommendation:** 3
**Confidence:** 4

**Summary:**

This paper proposes a novel evaluation metric called the reliability score for conditional generative models. Rather than assessing a single generated output (as done by metrics like CLIP score), the authors define reliability as the worst-case value of a user-specified similarity metric within a conformal prediction set at confidence level.

**Compliance With Llm Reviewing Policy:**

Affirmed.

**Key Questions For Authors:**

See weaknesses.

**Limitations:**

The following should be addressed more explicitly: (1) the marginal rather than conditional nature of coverage guarantees; (2) sensitivity to LGM quality and hyperparameter tuning.

**Strengths And Weaknesses:**

**Strengths:**

This well-written paper proposes a novel, flexible, and practically implementable metric for evaluating the reliability of generative models under uncertainty. By uniquely integrating latent-space conformal calibration with metric-aware worst-case optimization, the authors provide a tractable, convex solution that achieves massive computational speedups over existing methods.

**Weaknesses:**

1. Concerns still exist with respect to the reproducibility and the open-source nature of the evaluation tools in this paper. If the code or evaluation tools cannot be made open-source, it would significantly undermine the contribution of this work.
2. Theorem 3.3 relies on the assumption that the LGM *exactly* recovers the conditional distribution Dec(E(Ŷ,x), x) =ᵈ Ŷ|X=x. In practice this will only approximately hold, and the paper provides no analysis or empirical characterization of the sensitivity to this approximation error. How does coverage degrade when the VAE reconstruction is imperfect?
3. The text-to-image comparison involves only three models and a single metric (CLIP-SIM). The image-to-text experiments evaluate four models but the re-ranking results in Table 2 are somewhat modest and interpretation is partly post-hoc.

---

> ### Author Rebuttal · Authors · 2026-03-30
>
> We greatly appreciate the valuable suggestions and comments on our paper. We will provide responses to your questions below.
>
> ---
>
> **Question.** Reproducibility and open-source availability.
>
> **Response.** We will release the code, implementation details, and evaluation protocols upon acceptance.
>
> ---
>
> **Question.** About the discussion and sensitivity analysis of assumptions in Theorem 3.3.
>
> **Response.** Our condition can be relaxed: the decoder only needs to approximately recover the conditional distribution. Since $P(Dec(\mathcal{E}\_{n+1}(\widehat{Y}\_{n+1},X\_{n+1})) \in C_{\mathcal{Y}}(X_{n+1})) \geq 1-\alpha$, if $Dec(\mathcal{E}(\widehat{Y},x), x) \approx^d \widehat{Y}|X=x$, then $P(\widehat{Y}\_{n+1}\in C\_{\mathcal{Y}}(X\_{n+1})) \geq 1-\alpha$ holds approximately.
>
> We validate this via a sensitivity analysis on synthetic data ($\alpha=0.1$), varying LGM training epochs to track validation loss $\mathcal{L}$, empirical coverage in $\mathcal{Y}$, and prediction set area.
> As shown below, coverage is stable when $\mathcal{L}$ is small, but fails when large (Epochs 1-5). This suggests that the assumption is practically meaningful rather than merely technical.
> | Epoch      | 1      | 5      | 10     | 100    | 183    | 300    |
> |------------|--------|--------|--------|--------|--------|--------|
> | $\mathcal{L}$ | 0.3740 | 0.0197 | 0.0112 | 0.0086 | 0.0080 | 0.0084 |
> | Coverage in $\mathcal{Y}$        | 0.0205 | 0.8468 | 0.8830 | 0.8972 | 0.8915 | 0.8955 |
> | Area in $\mathcal{Y}$  | 23.24  | 287.12 | 278.39 | 235.49 | 232.70 | 246.78 |
>
> ---
>
> **Question.** Additional metrics and models in the  text-to-image experiment.
>
> **Response.** To provide a more comprehensive evaluation, we expanded our text-to-image experiments by adding Kandinsky-2.2 and a new structural metric, DINO-SIM ($\alpha=0.1$) that measures the structural similarity between generated and ground-truth images using DINOv2-base [1]  features. The evaluations across all models are shown below:
> | Model | DINO | CReL-DINO  |
> | :--- | :--- | :--- |
> | SD3-M | 0.4615 | -0.1480 |
> | SD3.5-L | 0.4531 | -0.1365 |
> | FLUX.1-dev | 0.4395 | -0.1411 |
> | Kandinsky-2.2 | 0.4407 | -0.1404 |
>
> For CLIP-SIM (num_z0=50), Kandinsky-2.2 averages 0.2603 but drops to 0.0062 under CReL's worst-case optimization.
>
> CReL reveals hidden flaws: SD3-M achieves the highest average DINO-SIM but the worst CReL-DINO score.  This proves that while SD3-M performs well on average, its structural generation is highly brittle in worst-case scenarios compared to SD3.5-L.
> We will include these results in the revision.
>
> [1] Oquab M, Darcet T, Moutakanni T, et al. Dinov2: Learning robust visual features without supervision[J]. arXiv preprint arXiv:2304.07193, 2023.
>
> ---
>
> **Question.** About the interpretation of our results in Tab 2.
>
> **Response.** The goal of Tab 2 is not necessarily to show a significantly different ranking result, but to demonstrate that reliability-aware evaluation can lead to conclusions different from standard average-case metrics. In fact, Tab 2 already shows meaningful ranking changes under both CLIP-SIM and BERT-SIM; for example, BLIP-base moves from last under CLIP to first under CReL-CLIP, and the gap between BLIP-base and BLIP-large is substantially enlarged under CReL-BERT.
>
> We also respectfully disagree that the interpretation is merely post-hoc. The manuscript does not rely only on isolated qualitative examples; rather, it explicitly explains the reranking through the **score distributions** shown in Fig 3. For example, the higher CReL-CLIP score of BLIP-base is attributed to its more concentrated score distribution relative to GIT-large, and the enlarged gap in CReL-BERT between BLIP-base and BLIP-large is explained in the same way.
>
> ---
>
> **Question.** Marginal rather than conditional nature of coverage guarantees.
>
> **Response.** The marginal, rather than conditional, nature of the coverage guarantee is standard in distribution-free conformal prediction and is not specific to our method; the same type of guarantee is also adopted in PCP (Wang et al., 2022) and Feldman et al. (2023). Exact conditional coverage is generally unattainable without imposing substantially stronger modeling assumptions. Our method, therefore, follows the standard conformal framework, while contributing a prediction-set construction that is both more informative and more computationally tractable under the same validity guarantee.
> - Wang, Z., Gao, R., Yin, M., Zhou, M., & Blei, D. (2023, April). Probabilistic Conformal Prediction Using Conditional Random Samples. In International Conference on Artificial Intelligence and Statistics (pp. 8814-8836). PMLR.
>
> ---
>
> **Question.** Sensitivity analysis of LGM.
>
> **Response.** We provide sensitivity analyses for the VAE's $\beta$ (Tab 4), SD's inference steps $T$ (Tabs 5-6), and optimization initial points (Figs 6-7). We also include the sensitivity analysis on validation loss for synthetic data (see Q2 above).

---

> > ### Author Rebuttal · Reviewer_w6et · 2026-04-02
> >
> > The author has addressed my concerns.

---

> > > ### Author Response · Authors · 2026-04-03
> > >
> > > Thank you for your follow-up. Since you indicated that our response has adequately addressed your concerns, we would respectfully ask you to reconsider your score so that it is consistent with your updated assessment.

---

### Official Review · Reviewer_qqAF · 2026-03-12

**Soundness:** 3
**Presentation:** 3
**Significance:** 3
**Originality:** 3
**Overall Recommendation:** 5
**Confidence:** 2

**Summary:**

This paper focuses on computing the worst-case value of a user-specified metric for conditional generative models at confidence level $1-\alpha$. They propose to identify the quantile region in the latent space, which reduces the computation cost of performing directional quantile regression (DQR) and enables optimizing the worst-case value using linear programming. Numerical results show the proposed CLIP score variant can effectively identify misalignments.

**Compliance With Llm Reviewing Policy:**

Affirmed.

**Final Justification:**

The author's response addresses my questions. I maintain my recommendation for acceptance.

**Key Questions For Authors:**

1. In Table 1, the prediction set region of DQR is much larger. Is there any explanation?
2. If two methods have comparable coverage, why is the prediction set region the smaller the better? Wouldn't it be better to identify the worst case for a larger region?

**Limitations:**

yes

**Strengths And Weaknesses:**

* **Soundness:** Identifying the feasible region in the latent space to reduce computation costs is technically sound. The experiments are well designed, and the results demonstrate the advantage of the proposed CReL-CLIP metric over the original CLIP score.
* **Presentation:** This paper is easy to follow. I do not check whether all the important references are included, as I am not an expert in this field.
* **Significance:** The proposed method aims to evaluate the reliability of conditional generative models. This paper not only makes improvements over the work of (Feldman et al., 2023) but also extends it to image-to-text/text-to-image models.

---

> ### Author Rebuttal · Authors · 2026-03-30
>
> We greatly appreciate the valuable suggestions and comments on our paper. We will provide responses to your questions below.
>
> ---
>
> **Question.** In Table 1, the prediction set region of DQR is much larger. Is there any explanation?
>
> **Response.** Thank you for your question. Compared to our method, the set of DQR has a larger size because it performs calibration separately for each dimension, which can be overly conservative. In contrast, we perform (step 3) calibration for the region jointly as a whole.
>
> ---
>
> **Question.** If two methods have comparable coverage, why is the prediction set region the smaller the better? Wouldn't it be better to identify the worst case for a larger region?
>
> **Response.** Thank you for your question. In conformal prediction, coverage is only one part of the goal. The other part is efficiency or informativeness. If two methods both achieve the desired coverage, then the one with the smaller prediction set is preferred because it gives a sharper description of uncertainty.
>
> A prediction set is supposed to represent the set of plausible outcomes. If that set is unnecessarily large, then it is valid but overly conservative: it covers the truth partly because it includes many values that are not truly plausible. Such a set provides less useful information to the user. By contrast, a smaller set with the same coverage means the method is localizing the plausible outcomes more precisely.

---

> > ### Author Rebuttal · Reviewer_qqAF · 2026-04-01
> >
> > Thank you for addressing my questions. I maintain my recommendation for acceptance.

---

> > > ### Author Response · Authors · 2026-04-02
> > >
> > > Dear Reviewer qqAF,
> > >
> > > Thank you for reviewing our rebuttal and confirming that your concerns have been resolved. We deeply appreciate your insightful feedback throughout the review process, which has meaningfully contributed to improving our manuscript.

---

### Official Review · Reviewer_F92m · 2026-03-12

**Soundness:** 3
**Presentation:** 4
**Significance:** 3
**Originality:** 3
**Overall Recommendation:** 4
**Confidence:** 4

**Summary:**

The paper introduces a reliability score for evaluating conditional generative models that captures the worst-case performance within a prediction set at a specified confidence level. To address the challenges of high-dimensional outputs and nonconvex similarity metrics, the authors propose Conformal ReLiability (CReL). The method first maps outputs into a learned latent space, constructs a Directional Quantile Region (DQR), and then applies conformal calibration to enforce marginal coverage. Then it computes the worst-case evaluation metric over the resulting convex calibrated set. The papers use text-to-image and image-to-text models to validate their methods.

**Compliance With Llm Reviewing Policy:**

Affirmed.

**Final Justification:**

The rebuttal has addressed my questions on computational cost, optimization stability, and the role of the latent generative model, and I will maintain my positive score.

**Key Questions For Authors:**

- What time and computational resources are required to train the latent generative model?

- You mentioned that the linear programming step is executed multiple times. How sensitive is the projected gradient descent procedure to initialization?

**Limitations:**

The authors did not discuss the limitations of their work. I believe one of the main limitations is the need to train the latent generative models for evaluation. This can cause biases and is computationally expensive for users to use this evaluation metric.

**Strengths And Weaknesses:**

The paper tackles an important question of evaluation for conditional generation.

Latent-space formulation improves tractability and scalability for high-dimensional outputs.

Empirical results show improved calibration and more meaningful evaluation compared to standard metrics (CLIP, BERT).

Weaknesses:
Overall, I am positive about this paper; however, my primary concerns regarding this paper are its reliance on training a latent generative model. While the paper provides theoretical guarantees, the theoretical contribution itself is limited, and the core idea is not novel. Furthermore, the need to train a latent generative model for every model is computationally expensive.

---

> ### Author Rebuttal · Authors · 2026-03-30
>
> We greatly appreciate the valuable suggestions and comments on our paper. We will provide responses to your questions below.
>
> ---
>
> **Question.** About the theoretical contribution. "While the paper provides theoretical guarantees, the theoretical contribution itself is limited, and the core idea is not novel."
>
> **Response.** Thank you for your question. The novelty of our paper lies primarily in introducing a new reliability metric for conditional generation, together with a principled latent-space conformal framework for computing it. The theoretical results are intended to provide the formal support for this framework.
>
> In particular, our method requires the prediction set to be both valid and informative, while also ensuring that the resulting reliability computation is tractable.
>
> To this end, Proposition 3.2 establishes coverage in the latent space after calibration, Theorem 3.3 shows that the decoded prediction set inherits the desired coverage guarantee, and Proposition 3.5 identifies the convexity and compactness structure that enables projected optimization with convergence guarantees.
>
> ---
>
> **Question.** About the sensitivity of initialization. "How sensitive is the projected gradient descent procedure to initialization...?
>
> **Response.** Thank you for this comment. We conduct a sensitivity analysis of the image-to-text task by evaluating the BLIP-base model using the CLIP-SIM score across ten runs with different random seeds. The resulting standard deviation is 0.00027, indicating that the metric is highly stable across random initializations.
>
> ---
>
> **Question.** About the computational cost of training a latent generative model. "Furthermore, the need to train a latent generative model for every model is computationally expensive."
>
> **Response.** We introduce the latent generative model to facilitate the optimization required for computing the reliability score. Moreover, we would like to emphasize that the latent-space formulation is adopted precisely to improve computational efficiency. As discussed in lines 194–211, directly performing calibration in the output space can be substantially more expensive. As demonstrated in Figure 8, our method is considerably more efficient than Feldman’s method, which performs calibration directly in the output space.
>
> To quantify the actual overhead, we report the training cost of our VAE (LGM) using an NVIDIA GeForce RTX 4090 48GB GPU with a batch size of 512.
> - For the synthetic dataset (MSE), the 385-epoch training takes only 2.80 minutes and consumes a peak VRAM of 0.06 GB.
> - For the higher-dimensional image-to-text task (BLIP-base, CLIP-SIM), the 484-epoch training takes just 15.11 minutes with a peak VRAM of 0.22 GB.
>
> These low time and memory requirements demonstrate that training the LGM is computationally inexpensive.  Moreover, this cost is incurred only once for each target generative model as an offline preprocessing step, after which the calibrated latent-space pipeline can be reused to evaluate many test examples.
>
> ---
>
> **Question.** About the validity of training LGM. "Training the latent generative models can cause biases..."
>
> **Response.** As claimed in Theorem 3.3, our calibration is valid as long as the decoder of LGM can well recover the conditional distribution. This can be achieved by many current LGMs, such as VAE and the stable diffusion model. To demonstrate, we conducted a sensitivity analysis in the simulation setting ($\alpha=0.1$), tracking validation loss $\mathcal{L}$, empirical coverage, and prediction set area across different training epochs.
>
> | Epoch      | 1      | 5      | 10     | 100    | 183    | 300    |
> |------------|--------|--------|--------|--------|--------|--------|
> | $\mathcal{L}$ | 0.3740 | 0.0197 | 0.0112 | 0.0086 | 0.0080 | 0.0084 |
> | Coverage in $\mathcal{Y}$        | 0.0205 | 0.8468 | 0.8830 | 0.8972 | 0.8915 | 0.8955 |
> | Area in $\mathcal{Y}$  | 23.24  | 287.12 | 278.39 | 235.49 | 232.70 | 246.78 |
>
> As shown above, coverage remains highly stable and valid as long as the LGM achieves a sufficiently low validation loss ($\ge$ Epoch 10). It only fails when the error is large (Epochs 1-5). This confirms that a properly trained LGM effectively mitigates the risk of evaluation bias.

---

> > ### Author Rebuttal · Reviewer_F92m · 2026-04-05
> >
> > I would like to thank the authors for the detailed rebuttal and for addressing my questions on computational cost, optimization stability, and the role of the latent generative model.
> >
> > The clarifications are helpful, but my main concerns have been resolved, and I will maintain my positive score.

---

> > > ### Author Response · Authors · 2026-04-06
> > >
> > > Dear Reviewer F92m,
> > >
> > >
> > >
> > > We sincerely appreciate you taking the time to read our rebuttal and confirming the resolution of your questions. Thank you for your valuable suggestions and for helping us improve the overall quality of our work.

---

### Official Review · Reviewer_Corf · 2026-03-13

**Soundness:** 3
**Presentation:** 3
**Significance:** 2
**Originality:** 2
**Overall Recommendation:** 4
**Confidence:** 4

**Summary:**

The paper proposes a reliability framework for assessing the capability and performance of generative models through conformal prediction sets. The main motivation is that complex outputs often lie in high-dimensional spaces, such as images or text, where standard metrics like Euclidean distance can be uninformative, computationally intractable, or overly conservative. To address this challenge, the paper develops a conformal prediction framework in a learned latent space rather than the original data space, with the goal of constructing valid prediction sets for complex high-dimensional outputs. The paper also provides theoretical guarantees for the validity of the resulting conformal sets in the original data space. The proposed method is evaluated on several modern real-data tasks and demonstrates its empirical effectiveness.

**Compliance With Llm Reviewing Policy:**

Affirmed.

**Final Justification:**

Overall, the paper has a clear and complete idea, and it is well presented and well written. Although I view its originality and novelty as somewhat limited, I still maintain my positive evaluation. In my view, the paper makes a meaningful contribution to the conformal prediction literature by considering the reliability score, and it is worthy of acceptance, even if the main technical advance is more incremental than fundamental.

**Key Questions For Authors:**

Is the validity of the proposed framework specific to DQR, or would it also hold for other multivariate conformal prediction methods once the latent embedding is constructed? If other methods also apply, a comparison would help justify the choice of DQR.

**Limitations:**

The paper would benefit from a more explicit discussion of its limitations. In particular, it should better clarify how strongly the method depends on the quality of the learned latent representation and decoder wrt assumption in Theorem 3.3, and how violations of these assumptions may affect coverage validity in the original data space.

**Strengths And Weaknesses:**

I like the motivation of the paper. Conformal prediction for high-dimensional outputs is an important and challenging problem, especially for modern tasks involving images, text, and other complex structured data. The paper is generally well motivated, the presentation is clear, and the work is complete: it includes theoretical development, empirical results, and comprehensive ablation studies, e.g., the computational cost of the proposed method compared to the baseline.

At the same time, I feel the theoretical and methodological novelty is somewhat limited,

**Strengths**

The paper addresses a practically important problem: applying conformal prediction to high-dimensional generative outputs where standard distances in the original space are difficult to use meaningfully. The work is fairly complete. It includes a clear motivation, a reasonably developed theoretical section, comprehensive experiments, and additional ablation studies. Overall, the paper appears carefully written.

**Weaknesses**

* From a theoretical and methodological perspective, the contribution feels somewhat limited. Once a meaningful latent representation is constructed, it is not fully clear how much of the contribution comes from the specific conformal method used here, namely DQR, rather than from the more general idea of applying conformal prediction in latent space. This raises an important question: does the validity argument or overall framework extend to essentially any multivariate conformal prediction method once a suitable latent representation is available? If so, the main novelty may lie more in the latent-space formulation than in the specific conformal procedure itself.

* The paper would also benefit from a stronger discussion of related work on conformal prediction for generative models. In particular, PCP by Wang et al. (2022) and subsequent follow-up works seem especially relevant, as they are more directly designed for generative settings in an end-to-end manner. A clearer empirical and conceptual comparison with this line of work would help better position the contribution.

* The assumptions used in Theorem 3.3 are acceptable to me, although somewhat restrictive. However, it would be helpful to include a sensitivity analysis examining how violations of these assumptions affect coverage validity in the original data space. In particular, I would like to better understand how the coverage validity behaves when decoder performance deteriorates or when the decoded conditional distribution deviates from the assumption. Such an analysis would make the practical robustness of the method much clearer.

If these issues are addressed more convincingly, I would be happy to increase my rating.

Wang, Z., Gao, R., Yin, M., Zhou, M., & Blei, D. (2023, April). Probabilistic Conformal Prediction Using Conditional Random Samples. In International Conference on Artificial Intelligence and Statistics (pp. 8814-8836). PMLR.

---

> ### Author Rebuttal · Authors · 2026-03-30
>
> We greatly appreciate the valuable suggestions and comments on our paper. We will provide responses to your questions below.
>
> ---
>
> **Question.** About methodological contributions beyond latent-space formulation. "It is not fully clear how much of the contribution comes from the specific conformal method used here, namely DQR..."
>
> **Response.**  Thank you for this question, which gives us the opportunity to further clarify our contribution. We propose DQR+calibration (step 3) for constructing the prediction set. The objective (lines 155-157, left) for constructing the prediction set is not merely to ensure validity, but to achieve **informative** calibration, which is especially important in high-dimensional conformal prediction because of the curse of dimensionality. Concretely, beyond satisfying the desired coverage guarantee, we seek prediction sets that are as small as possible.
>
> From this perspective, traditional calibration strategies such as naive multivariate quantile regression—which calibrates each coordinate separately and therefore yields rectangular prediction sets—or using DQR alone (lines 148–149, left column) are not fully satisfactory, as they can be overly conservative. To obtain more informative prediction sets, we introduce an additional calibration step following DQR. Importantly, this step calibrates the prediction region **jointly as a whole (equation 4)**, rather than dimension by dimension. As shown in Table 1, although both DQR and our method attain the required coverage guarantee, our approach produces substantially smaller prediction sets, demonstrating that the proposed calibration is considerably more informative.
>
> ---
>
> **Question.** Discussion and empirical comparison with related work. "In particular, PCP by Wang et al. (2022) and subsequent follow-up works seem especially relevant, as they are more directly designed for generative settings in an end-to-end manner..."
>
> **Response.** Thank you for your suggestion. PCP by Wang et al. (2022) also constructs the prediction set for conditional generative models. However, it does not consider the goal of computing the reliability score, which imposes requirements on the constructed prediction set for tractable optimization. For prediction set construction, PCP calibrates each dimension separately, which may lead to more conservative prediction sets than those produced by our joint calibration procedure.
>
> To demonstrate this, we conducted an empirical comparison with PCP on our non-linear simulation dataset with a target miscoverage rate of $\alpha=0.1$. For our CReL framework, the LGM is implemented as a VAE. For PCP, we utilize its default multi-dimensional setting and report the average results over 5 independent runs. Both methods (approximately) achieve valid coverage (ours: 0.8915; PCP: 0.9001), while  CReL produces a much tighter prediction set (PCP: 854.24; ours: 232.70), highlighting the ability of our joint calibration to produce more informative calibration than the dimension-wise calibration.
>
> We will include this discussion and comparison in the revised version.
>
> - Wang, Z., Gao, R., Yin, M., Zhou, M., & Blei, D. (2023, April). Probabilistic Conformal Prediction Using Conditional Random Samples. In International Conference on Artificial Intelligence and Statistics (pp. 8814-8836). PMLR.
>
> ---
>
> **Question.** About the discussion and sensitivity analysis of assumptions in Theorem 3.3. "The assumptions used in Theorem 3.3 are acceptable to me, although somewhat restrictive. However, it would be helpful to include a sensitivity analysis..."
>
> **Response.** Thank you for this question. Following your suggestion, we conduct a sensitivity analysis on synthetic data to examine how the coverage rate varies with the reconstruction error. Specifically, we vary the number of training epochs and report the corresponding  VAE loss $\mathcal{L}$, coverage in $\mathcal{Y}$, and area ($\alpha=0.1$).
>
> As shown below, the coverage remains stable when the $\mathcal{L}$ is sufficiently small, but fails when the error becomes large, such as at epochs 1 and 5. This suggests that the assumption is practically meaningful rather than merely technical. Moreover, many modern high-quality generative models, such as Stable Diffusion, achieve very small losses in practice, making this condition plausible in realistic applications.
> | Epoch      | 1      | 5      | 10     | 100    | 183    | 300    |
> |------------|--------|--------|--------|--------|--------|--------|
> | $\mathcal{L}$ | 0.3740 | 0.0197 | 0.0112 | 0.0086 | 0.0080 | 0.0084 |
> | Coverage in $\mathcal{Y}$        | 0.0205 | 0.8468 | 0.8830 | 0.8972 | 0.8915 | 0.8955 |
> | Area in $\mathcal{Y}$  | 23.24  | 287.12 | 278.39 | 235.49 | 232.70 | 246.78 |

---

> > ### Author Rebuttal · Reviewer_Corf · 2026-04-02
> >
> > I thank the authors for their response and further clarification. I will maintain my score.

---

> > > ### Author Response · Authors · 2026-04-03
> > >
> > > Thank you for your follow-up and for acknowledging that your concerns have been fully resolved. In your earlier comment, you noted that if these issues were addressed more convincingly, you would be happy to increase your rating. Since you now indicate that our response has adequately addressed the concerns, we would respectfully ask you to reconsider your score so that it is consistent with your updated assessment.

---

### Decision · Program_Chairs · 2026-04-30

**Decision:**

Accept (regular)

**Comment:**

The reviewers agree that this paper studies an important problem in evaluating conditional generative models beyond average-case metrics, and they find the submission clear and supported by a reasonably thorough empirical study. The strengths highlighted across reviews include the introduction of a reliability-oriented evaluation criterion based on conformal prediction, the use of a latent-space formulation to make the resulting optimization tractable, and consistently positive empirical results. At the same time, two reviewers raised concerns regarding the level of technical novelty, noting that parts of the conformal framework and theoretical analysis build on existing tools.

Based on the rebuttal and subsequent discussion, I find that the authors have provided sufficient clarifications and additional evidence addressing these concerns. The added comparisons to related conformal approaches, the sensitivity analysis, and the additional experiments in the text-to-image setting strengthen the empirical and practical aspects of the work. Reviewers indicated that their concerns were addressed after the rebuttal. Taking these inputs into account, I view the contribution as a useful and well-executed addition to the literature.